# TMMLU+: An Improved Traditional Chinese Evaluation Suite for Foundation Models

**Zhi Rui Tam**[*][†]**, Ya-Ting Pai**[*][†]**, Yen-Wei Lee, Jun-Da Chen, Wei-Min Chu, Sega Cheng**
iKala AI Lab
{avery.pai, patrick.lee}@ikala.ai

**Hong-Han Shuai**
National Yang Ming Chiao Tung University

## Abstract

We present TMMLU+, a new benchmark designed for Traditional Chinese language understanding. TMMLU+ is a multi-choice question-answering dataset with 66 subjects from elementary to professional level. It is six times larger and boasts a more balanced subject distribution than its predecessor, Taiwan Massive Multitask Language Understanding (TMMLU). We also benchmark closed-source models and 23 open-weight Chinese large language models (LLMs) of parameters ranging from 1.8B to 72B on the proposed TMMLU+. Our findings reveal that (1.) Traditional Chinese models still trail behind their Simplified Chinese counterparts, highlighting a need for more focused advancements in LLMs catering to Traditional Chinese. (2.) Current LLMs still fall short of human performance in average scores, indicating a potential need for future research to delve deeper into social science and humanities subjects. (3.) Among all the tokenization compression metrics examined, we identify that only the fertility score uniquely demonstrates strong correlations with our benchmark results. We foresee that TMMLU+ will pinpoint areas for future model improvement, thereby narrowing the gap between machine and human linguistic capabilities and supporting researchers in developing Traditional Chinese LLMs. Our dataset, along with the benchmark source code, is accessible at huggingface.co/datasets/ikala/tmmluplus.

## 1 Introduction

LLMs are a group of transformer models, typically trained in the autoregressive method. They can be used to solve a wide range of tasks in Natural Language Understanding (NLU) (Rajpurkar et al., 2016; Wang et al., 2019), vision processing (Goyal et al., 2017; Fu et al., 2023), and robotics (Padalkar et al., 2023). As the abilities of LLMs continue to improve, existing benchmarks start to saturate and supersede the human baseline. For example, Massive Multitask Language Understanding (MMLU) (Hendrycks et al., 2021a) is created to test knowledge understanding across a wide range of topics related to English and US culture. Since then, many subsequent works (Dac Lai et al., 2023; Son et al., 2024; Yin & Kawahara, 2024) have tried to extend the same multi-choice question-answering format with a wide variety of topics into other languages.

Recent efforts have sought to adapt this approach to the Chinese language. For example, as outlined in Huang et al. (2023); Li et al. (2023), the primary focus is on Simplified Chinese and the cultural milieu of mainland China. However, the Traditional Chinese domain, which is used by 30M population around the world, is nowhere to be found. Given that there are stark differences between these 2 languages despite sharing the same root. A primary distinction between Simplified and Traditional Chinese lies in their divergent

---

[*]Equal contribution
[†]Work performed while at iKala AI Lab

vocabularies. For instance, the concept of "computer memory" is expressed as "内存" in Simplified Chinese, while in Traditional Chinese it is "記憶體". A direct character-by-character conversion of the Traditional term to Simplified Chinese would yield "记忆体", which occupies an entirely different semantic space. This lexical divergence is further complicated by cases where identical characters carry disparate meanings across the two writing systems. A salient example is the character "丑", which in Simplified Chinese can denote either "ugly" or the period from 1 AM to 3 AM, depending on context. In contrast, Traditional Chinese employs "醜" to convey "ugly," while reserving "丑" exclusively for the aforementioned time designation.

Hsu et al. (2023) proposes TC-Eval, which features TMMLU, a Traditional Chinese multi-choice question-answering dataset. TMMLU covers 55 subjects and includes a total of 3,300 questions. Despite its advancements, it presents several limitations: Firstly, its scope is relatively narrow, with only 55 subjects and some featuring less than 100 questions, leading to potential gaps in subject diversity and coverage. Secondly, there are inconsistencies in the question formats, with 5% of questions deviating from the multi-choice format and 1% including more than four options. Lastly, the absence of a dedicated development set restricts the exploration of few-shot prompting techniques. Addressing these issues is crucial for advancing LLMs' understanding and applicability in linguistically and culturally diverse contexts.

We propose TMMLU+, an enhanced version of TMMLU, which aims to address the issues highlighted above. TMMLU+ contains 22,690 questions across 66 subjects, ranging from primary, secondary, undergraduate, and professional levels of education. Similar to Hendrycks et al. (2021a); Huang et al. (2023), we include STEM subjects: physics, chemistry, and computer science with mentions of Traditional Chinese culture. TMMLU+ benchmark specifically addresses the terminology differences between Traditional and Simplified Chinese, ensuring that models can accurately understand and respond to questions in the context of Taiwan. The terminology used in Traditional Chinese often differs from that in Simplified Chinese, reflecting variations in character forms, vocabulary, and usage. All subjects contain at least 110 questions, with 5 development questions for few-shot prompting.

We conduct experiments on 23 open-weight language models ranging from 1.8B to 72B from 5 different architectures of different pretraining datasets. To our surprise, Simplified Chinese language models outperform Traditional Chinese models in the same parameter level by at least 19%. The top-performing open-weight models are Qwen-72B and Qwen-14B, followed by Yi-34B-Chat model, with scores of 64.3%, 50.5%, and 49.6%, respectively. These models are predominantly used for Simplified Chinese. Despite the impressive results of these models, there is still a gap between their performance and that of human test-takers. On average, exam takers can score around 68% on the compared subjects. We have found that LLMs struggle in math-heavy subjects such as engineering math and accounting.

To dive deeper into the differences between Simplified Chinese language models and Traditional Chinese language models, we conduct translation experiments on TMMLU+ where we convert the Chinese characters (Traditional Chinese to Simplified Chinese) and repeat the evaluation on the newly converted version. We found that most Chinese language models perform better in simplified versions, with some showing a smaller margin than others. We also conduct experiments on few-shot prompting and chain-of-thought (CoT) prompting (Wei et al., 2022). We have found only few-shot prompting increases performance in most evaluated models while CoT prompting degrades. We also study open-weight Chinese language models from various architecture with different vocabulary sizes. Our analysis has found a strong correlation between the fertility scores and benchmark results.

## 2   Datasets

We introduce TMMLU+, a benchmark dataset to evaluate Traditional Chinese LLMs' multi-task language understanding capabilities.

## 2.1 Task overview

The dataset comprises 22,690 multi-choice questions from 66 subjects ranging from primary to professional level. These questions cover a wide range of subjects, including humanities, social science, and STEM fields. It includes universally applicable topics like physics, chemistry, and macroeconomics. Furthermore, some questions will require test-takers to choose the correct combination of options, such as 'ABCD', as their answer. In addition, we include questions where test-takers must provide comparison results, as shown in Figure 1.

下列電磁輻射的頻率大小順序排列何者正確？
(Which of the following is the correct order of electromagnetic radiation frequencies?)

  I: microwave

 II: $\gamma$ -rays

III: visible

IV: IR

 V: UV

A. I > IV > III > V > II

B. II > V > III > IV > I

C. V > II > IV > III > I

D. V > III > IV > II > I

Figure 1: Electromagnetic Radiation Frequency Order

臺東之所以成為臺灣極端高溫所在的原因，主要受颱風通過所造成的焚風所致。試問颱風中心在何處時，臺東最容易產生焚風，致使氣溫升高?
(Taitung experiences extremely high temperatures in Taiwan, primarily due to the foehn wind generated as typhoons pass by. At what location is the typhoon's center when Taitung is most likely to undergo foehn winds, leading to a temperature increase?)

A. 太平洋(Pacific Ocean)

B. 巴士海峽(Bashi Channel)

C. 臺灣海峽(Taiwan Strait)

D. 南海(South China Sea)

Figure 2: Taiwanese geography found in the Geography subject

Additionally, the dataset features questions requiring specific knowledge of Taiwan, sourced from standardized tests in subjects such as Taiwanese Hokkien, the culture of the aboriginal peoples of Taiwan, and the geography of Taiwan as shown in Figure 2. Questions from the Taiwanese Trivia Question Answering (TTQA) (Ennen et al., 2023) dataset are also included. TTQA is a multi-choice question evaluation about culture, geography, and history specific to Taiwan. To correctly answer these questions, a model needs to have an understanding of Taiwan culture. To view the list of included subjects, please refer to Appendix B.

## 2.2 Data collection

We have gathered previous exams between 1998 and 2023 from various public exams such as certificate exams, comprehensive assessment programs for junior students, and college entrance examinations. This results in 962 exams from different standardized test resources and PDF documents available online. In addition, we have also included subjects from TMMLU (Hsu et al., 2023) and TTQA (Ennen et al., 2023). Subjects related to culinary arts have been consolidated into a unified culinary category, and questions about dentistry have been incorporated into our preliminary set of questions. To ensure that the testing set contains more than 100 questions, we have manually added 29 more questions to the 103 questions from TTQA. We have named this new subject TTQAv2, as it is a distinct set of questions from TTQA.

We then eliminate questions containing images and transform mathematical and chemical equations into LaTeX format using GPT-4-vision (OpenAI, 2023) capabilities. Next, we use n-gram filtering (Broder, 1997) to remove similar questions within the same subject and eliminate redundancy. Lastly, we manually check all questions and remove any that require figures for answers or depend on the context of preceding questions. Once the

questions are finalized, we follow the same practice from Hendrycks et al. (2021a); Huang et al. (2023); Li et al. (2023) by randomly shuffling the choices of each question to ensure all subjects' answers are uniformly distributed. This results in a random guess accuracy of 25%. After refining the question pool, we establish human baseline scores for 44 of the 66 subjects. These baselines are instrumental for conducting a nuanced ablation study on human performance benchmarks in subsequent phases of our research.

## 2.3 Format

Each subject is then split into test, dev, and validation. We ensure that each subject contains at least 110 questions, with 5 development questions for few-shot prompting, and at least a minimum of 5 validation questions. Every question has four choices: A, B, C, or D, and the model is prompted to select one answer.

All 66 subjects can be grouped under 4 categories: STEM, Humanities, Social Science, and Others. For example, STEM subjects would test the arithmetic ability of LLMs, while humanities and social science focus more on the knowledge brevity of the models. Table 1 shows the statistic results for all categories.

| Category | Test | Dev | Validation |
|---|---|---|---|
| STEM | 3,458 | 70 | 385 |
| Social Sciences | 5,958 | 90 | 665 |
| Humanities | 1,763 | 35 | 197 |
| Other (Business, Health, Misc.) | 8,939 | 135 | 995 |
| Total | 20,118 | 330 | 2,242 |

Table 1: Data Categories and Counts

## 2.4 Evaluation

For all subjects, we calculate the average accuracy score first followed by averaging all subjects by their subject category. Finally, an average score is calculated based on the 4 categories' average scores.

## 2.5 Check for pretraining data leakage

Given the age of our questions, we suspect they may be present in the LLMs' training data, potentially compromising our goal to benchmark the LLMs' abilities. We use Min-K%++ (Zhang et al., 2024) which is the current SOTA in pretraining data detection for LLMs. We use Qwen-7B as the model and find that only 0.48% questions exist in its pretraining data.

# 3 Experiments

We conduct evaluations on 23 distinct LLMs, focusing on multilingual and Chinese-specific models. Each model varies in aspects such as tokenization, pretraining datasets, size of parameters, and architectural design. Furthermore, we compare the performance of these LLMs against datasets comprising human-generated answers to assess their relative effectiveness. Additionally, we investigate several factors that could influence the performances of LLMs on these datasets.

## 3.1 Prompting method

In our approach, we formulate the prompts for the LLMs in both zero-shot and few-shot settings. For the few-shot setting, we utilize the development splits, specifically employing the provided set of five examples as context examples. The complete prompts can be found in Appendix D.

| Model | Lang | STEM 0-shot | STEM 5-shot | Soc. Sci. 0-shot | Soc. Sci. 5-shot | Humanities 0-shot | Humanities 5-shot | Other 0-shot | Other 5-shot | Average 0-shot | Average 5-shot |
|---|---|---|---|---|---|---|---|---|---|---|---|
| **Random** | - | 25.0 | 25.0 | 25.0 | 25.0 | 25.0 | 25.0 | 25.0 | 25.0 | 25.0 | 25.0 |
| Qwen-72B | SC | 61.1 | 60.8 | 71.7 | **72.3** | 63.0 | 63.4 | 61.3 | **62.1** | 64.3 | 64.7 |
| GPT-4 | multi | 60.4 | **66.0** | 67.4 | **74.9** | 56.0 | **63.6** | 57.6 | **65.3** | 60.3 | **67.5** |
| Qwen-14B | SC | 46.9 | 46.4 | 56.7 | **58.1** | 49.4 | **51.5** | 48.8 | **49.4** | 50.5 | **51.4** |
| Gemini-pro | multi | 45.4 | 45.6 | 57.3 | **59.1** | 48.8 | **49.3** | 48.2 | **50.4** | 49.9 | **51.1** |
| Yi-34B-Chat | SC | **40.2** | 35.8 | **56.8** | 42.8 | **54.0** | 37.2 | **47.6** | 39.9 | **49.6** | 38.9 |
| Qwen1.5-14B-Chat | SC | 39.7 | **48.1** | 52.8 | **59.0** | 43.9 | **51.4** | 45.0 | **50.9** | 45.3 | **52.4** |
| Claude-1.3 | multi | 42.7 | **46.9** | 49.3 | **57.3** | 42.2 | **52.7** | 44.1 | **51.1** | 44.6 | **52.0** |
| GPT-3.5-turbo | multi | 41.6 | 42.8 | 46.7 | **51.2** | 36.7 | **41.2** | 42.0 | **46.3** | 41.8 | **45.3** |
| Claude-3.0 Opus | multi | 43.0 | **61.3** | 45.5 | **69.6** | 35.8 | **63.0** | 40.2 | **63.0** | 41.1 | **64.2** |
| Qwen-7B | SC | 37.5 | **38.4** | 45.5 | **48.3** | 38.1 | **42.3** | 39.0 | **42.0** | 40.0 | **52.0** |
| Yi-9B | SC | 28.7 | **44.7** | 44.6 | **59.0** | 48.5 | **49.7** | 36.8 | **49.7** | 39.6 | **59.8** |
| Qwen-1.8B | SC | **32.6** | 31.3 | **38.9** | 37.0 | **38.3** | 32.4 | 35.3 | 34.3 | **36.3** | 33.8 |
| Claude-2.0 | multi | **39.7** | 11.7 | **39.1** | 35.6 | 28.6 | 20.8 | **37.5** | 19.5 | **36.2** | 21.9 |
| Breeze-7B-Instruct-v1.0 | TC | **26.7** | 21.1 | **43.0** | 32.5 | **39.8** | 32.4 | **34.8** | 28.3 | **36.1** | 28.6 |
| ChatGLM3-6B | SC | 31.0 | 32.8 | 39.3 | 38.3 | 35.6 | 34.2 | 35.6 | 35.0 | 35.4 | 35.1 |
| Taiwan-LLM-13B | TC | 18.5 | **25.2** | 27.6 | 25.3 | 17.8 | **18.2** | 21.5 | 20.5 | 21.3 | **22.3** |
| Taiwan-LLM-7B | TC | 15.0 | 7.7 | 16.2 | 4.5 | 15.0 | 3.2 | 16.2 | 5.1 | **15.6** | 5.1 |

Table 2: Zero-shot/Five-shot average accuracy (%) in an answer-only setting. The average accuracy over the subjects within each category. SC: Simplified Chinese, TC: Traditional Chinese

## 3.2 Setup

For open-weight models, we use the chat template provided by each model if available and use the same prompting format (Huang et al., 2023) for non-conversation LLMs. Responses are parsed using a modified regular expression by Huang et al. (2023) with Traditional Chinese characters added. For close-source models, we obtain the response via APIs and parse using the same set of parsers used by open-weight models.

## 3.3 Models

To provide an extensive overview of the current landscape of Chinese LLMs, we conduct benchmarks not only on Traditional Chinese-specific models (Lin & Chen, 2023) but also on Simplified Chinese LLMs (Bai et al., 2023; Du et al., 2022). Our evaluation encompasses a range of well-regarded closed-source LLMs, including those from OpenAI GPT series (Brown et al., 2020), Gemini (Team, 2023), and Claude (Anthropic, 2023; 2024). Within the OpenAI GPT suite, our tests incorporate GPT-4 from the gpt-4-0613 model and GPT-3.5-turbo from the gpt-3.5-turbo-0613 model. For Gemini models, we use the Gemini-pro from the gemini-1.0-pro-001 model, the only publicly accessible Gemini model at the current point of writing. Regarding Claude models, we include Claude-1.3, Claude-2.0, and Claude-3.0 (Anthropic, 2024) in our evaluation where Claude-1.3 is solely trained in the supervised fine-tuned (SFT) (Lowe & Leike, 2022), whereas Claude-2.0 undergoes training in the SFT stage, followed by reinforcement learning from human feedback (Bai et al., 2022). The full models' detail description can be found in Appendix E.

## 3.4 Results

### 3.4.1 Comparative Analysis of Diverse Model Architectures

In our benchmark, we examine a spectrum of open-weight models, with parameter counts ranging from 1.8B to 72B (Bai et al., 2023; Lin & Chen, 2023; Ennen et al., 2023; Du et al., 2022), as well as proprietary models that are evaluated via API calls (OpenAI, 2023; Team, 2023; Anthropic, 2023; 2024). Table 2 presents our findings, which indicate a general trend where models with larger numbers of parameters tend to show improved performance across various domains. It is noteworthy that the open-source model, Qwen-72B, achieves superior zero-shot learning scores in comparison to the closed-source GPT-4 model, albeit GPT-4 exhibits enhanced performance in few-shot scenarios, surpassing Qwen-72B by a considerable margin.

| Model | Lang | STEM | Soc. Sci. | Humanities | Other | Average |
|---|---|---|---|---|---|---|
| **Human** | - | 62.9 | **72.1** | **69.5** | **68.5** | **68.2** |
| Qwen-72B | SC | **63.9** | 69.9 | 52.1 | 60.6 | 62.1 |
| GPT-4 | multi | 61.7 | 65.7 | 49.2 | 57.0 | 58.4 |
| Qwen-14B | SC | 48.3 | 54.8 | 40.3 | 46.4 | 47.5 |
| Gemini-pro | multi | 47.7 | 54.7 | 42.2 | 45.9 | 47.6 |
| GPT-3.5-turbo | multi | 43.6 | 43.8 | 31.5 | 40.3 | 29.8 |

Table 3: Zero-shot average accuracy (%) in 44 subjects with human answers data

Disparities in performance are noted even among models with comparable parameter sizes, which can be attributed to differing developmental approaches. Such discrepancies highlight the importance of model architecture and training regimen over the mere scale of parameters. We intend to further investigate these differences, focusing on how factors such as model tokenization contribute to the final accuracy in Section 3.5.4. Detailed comparisons across all models are available in Appendix F.

### 3.4.2 Human baseline

In this analysis, we compare human test-takers and a range of LLMs, including Qwen-72B, Qwen-14B, and the proprietary GPT-4, GPT-3.5-turbo, Gemini-Pro. The human baseline scores, as represented in Table 3, have been revised to reflect the passing rate based on the proportion of correct responses to total responses for each question. To illustrate, if a question offered four possible answers and the majority of human respondents selected the correct one, the passing rate for that question would be calculated by dividing the number of correct answers by the total number of responses.

Our result indicates that human test-takers maintain a lead over LLMs in zero-shot learning tasks, with an average accuracy of 68.2% across all domains. Notably, humans exhibit the highest proficiency in Social Sciences and Humanities, areas where nuanced understanding and contextual knowledge play significant roles. In comparison, the highest-performing model, Qwen-72B, shows an average accuracy of 62.1%, with particular strength in Social Science and STEM-related tasks.

### 3.4.3 Comparative Results Across Different Subjects

Based on results from Tables 2, we find subjects that necessitate mathematical reasoning, such as those in the STEM category, consistently show lower performance metrics compared to subjects that are more reliant on background knowledge and linguistic comprehension, such as Social Sciences, Humanities, and Other categories. This trend is evident in both prompting strategies, where models like GPT-4 and Qwen-72B while performing admirably across all categories, exhibit a noticeable dip in their performance in the STEM field.

### 3.5 Analysis and discussion

### 3.5.1 Comparative Analysis of Traditional and Simplified Chinese Characters

This study aims to investigate the impact of the linguistic differences between Traditional and Simplified Chinese characters on the performance of LLMs. Traditional and Simplified Chinese characters share only about 30% of their vocabulary, suggesting that LLMs trained in Traditional Chinese may encounter difficulties when processing Simplified Chinese text. Understanding the linguistic flexibility and limitations of LLMs within the context of these distinct yet interconnected writing systems is crucial. We utilized the widely recognized *opencc* (Kuo, 2023) tool to convert TMMLU+ questions and prompts from Traditional to Simplified Chinese. As detailed in Table 4, Traditional Chinese LLMs tend to perform better in the Traditional Chinese version of TMMLU+ in Social Sciences, which generally require a deep understanding of cultural context, and also show a slight advantage in other domains

| Model | Lang | STEM | | Soc. Sci. | | Humanities | | Other | | Average | |
|---|---|---|---|---|---|---|---|---|---|---|---|
| | - | SC | TC | SC | TC | SC | TC | SC | TC | SC | TC |
| GPT-3.5-turbo | multi | **42.1** | 41.6 | 46.3 | **46.7** | **37.6** | 36.7 | 41.7 | 42.0 | 41.9 | 41.8 |
| Gemini-pro | multi | **46.7** | 45.4 | **59.1** | 57.3 | **51.2** | 48.8 | **48.8** | 48.2 | **51.5** | 49.9 |
| Qwen-14B | SC | **48.1** | 46.9 | **58.0** | 56.7 | **49.8** | 49.4 | **49.3** | 48.8 | **51.3** | 50.5 |
| Qwen-7B | SC | **39.8** | 37.5 | **46.7** | 45.5 | **40.6** | 38.1 | **41.3** | 39.0 | **42.1** | 40.0 |
| Breeze-7B-Instruct-v1.0 | TC | **29.2** | 26.7 | 40.7 | **43.0** | 38.2 | **39.8** | **34.9** | 34.8 | 35.7 | **36.1** |
| Taiwan-LLaMa-13B | TC | 15.5 | **18.5** | 19.9 | **27.6** | 13.5 | **17.8** | 15.0 | **21.5** | 16.0 | **21.3** |
| Taiwan-LLaMa-7B | TC | 14.3 | **15.0** | 16.4 | 16.2 | **16.6** | 15.0 | 16.2 | 16.2 | 15.9 | 15.6 |

Table 4: Comparative Results on TMMLU+ in zero-shot settings: Simplified vs. Traditional Chinese

such as Humanities and STEM. Our findings indicate that using Traditional Chinese prompts can enhance the performance of LLMs trained specifically in Traditional Chinese, while Simplified Chinese prompts significantly improve the performance of multilingual and Simplified Chinese LLMs.

### 3.5.2 Language Models Performance on Simplified Chinese Benchmarks

In this section, we compare our dataset against established Simplified Chinese benchmarks, namely CMMLU (Li et al., 2023) and C-Eval (Huang et al., 2023), using a selection of 8 language models: Qwen series, both Traditional Chinese LLMs Breeze and Taiwan LLaMA as well as GPT-4. The comparison in Figure 3 revealed that the models tended to show a consistent level of accuracy, answering questions from both CMMLU and C-Eval with a similar accuracy. The figure highlights that all chinese language models generally performed better on the Simplified Chinese benchmarks, with a trend of results moving toward the upper left corner. Comprehensive outcomes for these models can be found in the Appendix G.

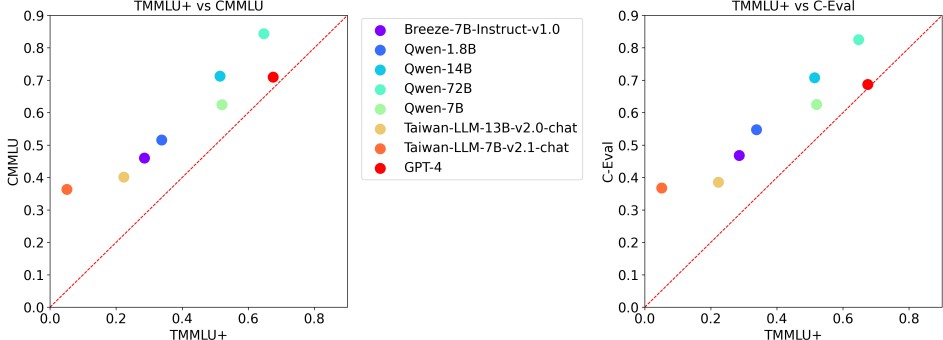

Figure 3: Comparison of Model Performance: TMMLU+ vs. C-Eval and CMMLU in five shot prompting

### 3.5.3 Does CoT improve performance on TMMLU+?

As observed in GPT-4 (OpenAI, 2023) and Gemini-Pro (Team, 2023), we use the few-shot CoT to improve the model reasoning ability in STEM subjects found in MMLU (Hendrycks et al., 2021a). We modify the prompt from a direct answer to the Chinese version of the CoT similar to Li et al. (2023) but using Traditional Chinese reasoning. All the thought processes are first generated by GPT-4 (OpenAI, 2023) and reviewed by humans. Table 5 shows performance degradation in the five-shot CoT evaluation setting which coincides with previous Chinese multi-choice question (Huang et al., 2023; Li et al., 2023) results.

| Model | STEM | | Soc. Sci. | | Humanities | | Other | | Average | |
|---|---|---|---|---|---|---|---|---|---|---|
| | w/o | CoT | w/o | CoT | w/o | CoT | w/o | CoT | w/o | CoT |
| Gemini-pro | 45.6 | **46.4** | 59.1 | 52.2 | 49.3 | 41.4 | 50.4 | 45.2 | 51.1 | 46.3 |
| Qwen-14B | 46.4 | 40.1 | 58.1 | 47.2 | 51.5 | 39.7 | 49.4 | 40.5 | 51.1 | 41.9 |
| GPT-3.5-turbo | 42.8 | 30.2 | 51.2 | 33.0 | 41.2 | 27.2 | 46.3 | 29.6 | 45.3 | 30.0 |

Table 5: Average accuracy (%) in a five-shot scenario with/without CoT setting. Scores highlighted in bold indicate enhanced performance relative to five-shot without CoT

| Model | Renyi Entropy | Average Token | Fertility | TMMLU+ Average |
|---|---|---|---|---|
| Yi-6B-Chat | 0.516 | 90.32 | 1.49 | 44.64 |
| Qwen-7B | 0.459 | 85.92 | 1.40 | 40.01 |
| Qwen-7B-Chat | 0.459 | 85.92 | 1.40 | 39.53 |
| BlueLM-7B-Base | 0.519 | 92.37 | 1.56 | 37.90 |
| Breeze-7B-Instruct-v1.0 | 0.453 | 79.10 | 1.30 | 36.10 |
| ChatGLM3-6B | 0.494 | 82.78 | 1.38 | 35.40 |
| DeepSeek-LLM-7B-Chat | 0.532 | 121.10 | 1.99 | 35.17 |
| BLOSSOM-v3.1-mistral-7b | 0.519 | 135.61 | 2.25 | 34.15 |
| Yayi-7b | 0.453 | 73.09 | 1.12 | 28.95 |
| DeepSeek-LLM-7B-Base | 0.532 | 121.10 | 1.99 | 26.31 |
| Mistral-7B-80k | 0.460 | 135.00 | 2.23 | 23.79 |
| Chinese-Llama-2-7b | 0.459 | 135.61 | 2.84 | 16.01 |
| Yayi-7B-Llama2 | 0.459 | 171.43 | 2.84 | 15.91 |
| Taiwan-LLM-7B-v2.1-chat | 0.459 | 135.61 | 2.84 | 15.61 |
| Atom-7B | 0.478 | 119.20 | 1.97 | 9.68 |
| Pearson Correlation | 0.366 | -0.685 | -0.742 | - |

Table 6: All 7B model zero-shot results for fertility score ablation study

We attribute this drop to insufficient relevance among bridging objects, detailed in Section 5 (Wang et al., 2023). As we go through each reasoning examples we found most errors contributed by false understanding of the world which leads to incorrect answer.

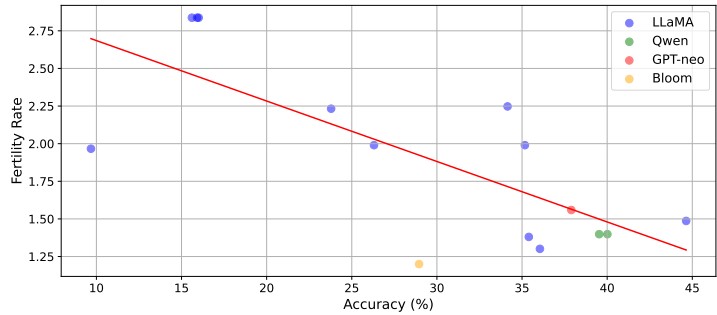

Figure 4: Fertility score and average TMMLU+ accuracy from 15 different LLMs around 7B parameters

### 3.5.4 Impact of Tokenizer Vocabulary

In our study, we identify a notable correlation between the size of a model's vocabulary and its performance in TMMLU+ tasks. Despite the broad range of pretraining corpora sizes, from 10 billion to 2 trillion tokens, which typically remain undisclosed, our findings highlight the significant impact of tokenizer design on model effectiveness. Specifically, in

the context of machine translation, we reference the study by (Zouhar et al., 2023), which demonstrates a strong correlation between Rényi entropy—a measure reflecting the diversity and distribution of tokens—and the BLEU score, a widely recognized metric for evaluating translation accuracy. Another work (Ali et al., 2023) has found that a low fertility score for tokenizers across different languages is crucial for multilingual performance. A tokenizer with a low fertility score, when presented with the same sentence, generates a shorter sequence of tokens compared to a tokenizer with a higher fertility score. In the case of Taiwan-LLaMA, 59.5% of Traditional Chinese characters in TMMLU+ are encoded with 256 bytes. This poses a challenge for the transformer module as it increases the input sequence while operating on a smaller vocab subset. To validate our hypothesis, we aggregate data from approximately 15 LLMs around 7B parameters in LLaMA (Touvron et al., 2023a;b; Jiang et al., 2023), Qwen (Bai et al., 2023), Bloom (Scao et al., 2022) and GPT-Neo (Black et al., 2022) architecture, and plot the correlation between their tokenizer fertility scores and average accuracy. Our result, as shown in Figure 4, reveals a significant Pearson correlation coefficient of -0.742 between fertility score and average accuracy, indicating that high fertility negatively impacts downstream performances.

### 3.5.5 Comparative Analysis of Error Patterns in LLMs and Human Test-Takers

In this section, we undertake a thorough examination of the error patterns exhibited by various LLMs in comparison to those made by human test-takers. Our goal is to discern whether the mistakes made by these advanced language models are akin to those typically encountered in human responses.

We partition the questions into three distinct categories based on their difficulty levels: **easy**, **medium**, and **hard**. The difficulty level of each question is determined through an empirical approach. First, we compute an average threshold for the correct response rates of humans across all questions. This threshold is used as a benchmark to categorize each question. Specifically, a question is classified as '*hard*' if its correct response rate is lower than or equal to 25%, '*medium*' if the rate is above 25% but lower than or equal to the average threshold, and '*easy*' if the rate is higher than the average threshold. Second, the categorized data is then used to populate arrays corresponding to each difficulty level, thereby enabling a comparative analysis of error patterns across the different categories. Finally, we analyze the distribution of errors in various categories and find that Qwen-72B outperforms humans in difficult questions across all major categories. However, most questions are dominated by easy difficulty, resulting in an average Qwen-72B score that falls short compared to humans.

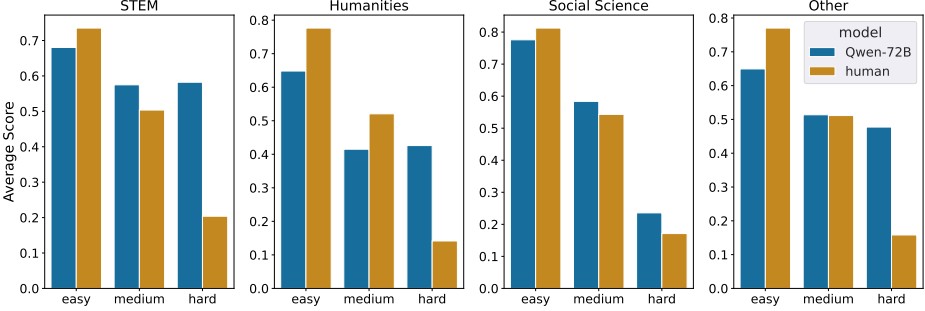

Figure 5: Human level difficulty comparison with Qwen-72B zero-shot settings.

### 3.5.6 Comparison of Scoring Methods

Existing evaluation libraries for multi-choice question answering, such as *lm-eval-harness* (Gao et al., 2023), typically measure the negative log-likelihood of choice terms and select the option with the highest log probability as the final answer. In contrast, our evaluation relies on string matching to extract the final answer, more closely mimicking real-world user interactions. However, this method can yield scores lower than random choice if the parser fails to extract a final answer. We compared these two methods across three evaluated

models, as shown in Table 7. Our findings indicate that the likelihood method generally produces higher scores. Given this discrepancy, we recommend that future comparisons of newer LLMs carefully consider and explicitly state the scoring method used to ensure fair and accurate benchmarking.

| Model | STEM | | Soc. Sci. | | Humanities | | Other | | Average | |
|---|---|---|---|---|---|---|---|---|---|---|
| | Likelihood | String | Likelihood | String | Likelihood | String | Likelihood | String | Likelihood | String |
| Breeze-7B-Instruct-v1.0 | **36.94** | 26.7 | **47.18** | 43.0 | 38.57 | **39.8** | **40.73** | 34.8 | **41.79** | 36.1 |
| Qwen-7B | 35.29 | **37.5** | 43.27 | **45.5** | 34.49 | **38.1** | **39.12** | 39.0 | 39.28 | **40.0** |
| Taiwan-LLM-7B | **26.37** | 15.0 | **30.26** | 16.2 | **26.21** | 15.0 | **28.88** | 16.2 | **28.62** | 15.6 |

Table 7: Comparison of Models Across Different Fields with Likelihood (Gao et al., 2023) and String Parsing (ours) results

## 4 Related Works

Many benchmark datasets have been created to evaluate the abilities of models in NLU tasks. Some of these benchmarks focus on specific subjects. For instance, the GLUE benchmark (Wang et al., 2019) is a collection of NLU tasks that mainly concentrate on language skills. AI2 Reasoning Challenge (Clark et al., 2018) is a dataset comprising multi-choice questions from science exams at various grade levels. On the other hand, MATH (Hendrycks et al., 2021b) is designed to evaluate LLMs' ability to solve mathematical problems. MMLU (Hendrycks et al., 2021a) is a benchmark used to assess models' knowledge comprehension in different domains. It provides broad and in-depth coverage of subjects, making it widely used for evaluating LLMs' abilities in knowledge understanding. The proliferation of benchmarks is increasing and expanding; however, it is noteworthy that most of these benchmarks have been developed in English.

With Chinese being the language having the largest user base worldwide, there also has been a significant increase in the number of benchmark datasets available in Chinese recently. For example, a bilingual benchmark including both English and Chinese has been released by Zhong et al. (2023) which focuses on standardized tests. Furthermore, some datasets are tailored to specific fields, such as CMATH (Wei et al., 2023), which evaluates the ability of LLMs to solve mathematical problems at the elementary level. Additionally, Zeng (2023) proposed MMCU, which covers four key areas: medicine, law, psychology, and education. Researchers have also created benchmarks that cover a wide range of subjects and difficulty levels from elementary school to college, such as M3KE (Liu et al., 2023), CMMLU (Li et al., 2023), and C-Eval (Huang et al., 2023).

However, considering the discrepancies in idiomatic usage and cultural nuances between Traditional Chinese and Simplified Chinese users, the benchmark datasets in Simplified Chinese might not be entirely suitable for accurately assessing LLMs' NLU capabilities in Traditional Chinese. Despite this scarcity, there are a few existing benchmarks specifically designed for Traditional Chinese. These include DRCD (Shao et al., 2019) focusing on reading comprehension, TTQA (Ennen et al., 2023) emphasizing Taiwanese-related knowledge extracted from Wikipedia pages. Also, Hsu et al. (2023) released a benchmark encompassing a diverse range of knowledge in 55 subjects from Taiwanese standardized examinations.

## 5 Conclusion

In this work, we introduce TMMLU+, an improved benchmark for evaluating large language models understanding in Traditional Chinese through multi-choice question answering. Our preliminary results have found that existing Traditional Chinese LLMs still fall behind their Simplified Chinese counterparts in terms of understanding Traditional Chinese. Furthermore, our analysis indicates that none of the current LLMs achieve an average score surpassing human performance, highlighting the challenging nature of our dataset and its potential to drive forward the development of more sophisticated language understanding systems.

## Acknowledgements

We would like to thank Cheng-Kuang Wu for his review and valuable feedback. We also thank Wen-Zhong Fu for adding our dataset to EleutherAI's lm-evaluation-harness tool.

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

# A TMMLU+ Overview

**Data Source**

1. Data was collected from different standardized test resources and PDF documents available online (90.88 %)

2. Taiwanese Trivia Question Answering (8.45 %)

3. Taiwan Massive Multitask Language Understanding (0.67 %)

**Collection Method(s) Used**

1. Artificially Generated

2. Scraped or Crawled

3. Manually parsed

**Data Processing**

1. Eliminate questions containing images and transform mathematical and chemical equations into LaTeX format using GPT-4-vision

2. Use n-gram filtering to remove similar questions within the same subject and eliminate redundancy

3. Manually check all questions and remove any that require figures for answers or depend on the context of preceding questions

**Data Format**

| Field Name | Field Value | Description |
|---|---|---|
| id | Integer | The order number of the data point |
| question | String | The multi-choice question |
| A | String | Choice A for the question |
| B | String | Choice B for the question |
| C | String | Choice C for the question |
| D | String | Choice D for the question |
| answer | String | The ground-truth answer of the question |

Table 8: Data Format

**Motivations**

1. Purpose: Research

2. Domain(s) of Application: Machine Learning, Natural Language Processing, Large Language Models (LLMs)

3. Motivating Factor(s): Narrow the gap between Traditional Chinese LLMs and Simplified Chinese LLMs development

# B Details of TMMLU+

**Subject details** Table 9, 10, 11, 12 shows each of the subjects found in TMMLU+ with their broad category, as well as total number of questions.

| Subject Name | Chinese Name | Questions |
|---|---|---|
| Engineering Math 11 | 工程數學 | 119 |
| Organic Chemistry 12 | 有機化學 | 126 |
| Advance Chemistry 13 | 化學 | 142 |
| Physics 14 | 物理 | 113 |
| Secondary Physics 15 | 高中物理 | 130 |
| Pharmacy 16 | 藥劑學 | 440 |
| Computer Science 17 | 資訊工程 | 198 |
| Basic Medical Science 18 | 基礎醫學 | 1065 |
| statistics and Machine Learning 19 | 統計與機器學習 | 254 |
| Junior Science Exam 20 | 國中會考基測自然科 | 242 |
| Junior Math Exam 21 | 國中會考基測數學科 | 199 |
| TVE Natural Sciences 22 | 統測自然科 | 476 |
| Junior Chemistry 23 | 國中理化 | 237 |
| TVE Mathematics 24 | 統測數學 | 172 |

Table 9: Overview of subjects in STEM

| Subject Name | Chinese Name | Questions |
|---|---|---|
| Clinical Psychology 25 | 臨床心理學 | 144 |
| Taiwanese Trivia Question Answering v2 (ttqav2) 26 | 台灣在地用語 | 131 |
| Human Behavior 27 | 人類行為與社會 | 348 |
| National Protection 28 | 軍事 | 240 |
| Politic Science 29 | 政治 | 1111 |
| Educational Psychology 30 | 教育心理 | 201 |
| Education (Profession Level) 31 | 教育專業 | 545 |
| Economics 32 | 經濟學 | 442 |
| Psychology Disorders 33 | 心理障礙職能治療學 | 608 |
| Geography of Taiwan 34 | 台灣地理 | 858 |
| Physical Education 35 | 體育 | 204 |
| Macroeconomics 36 | 總經 | 462 |
| Chinese Language and Literature 37 | 國文 | 226 |
| Junior Chinese Exam 38 | 國中會考基測國文 | 200 |
| TVE Chinese Language 39 | 統測國文 | 542 |
| Education 40 | 教育常識 | 143 |
| Three Principles of People 41 | 三民主義 | 160 |
| Taiwanese Hokkien 42 | 閩南語 | 148 |

Table 10: Overview of subjects in Social Science

| Subject Name | Chinese Name | Questions |
|---|---|---|
| General Principles of Law 43 | 法學大意 | 123 |
| Anti Money Laundering 44 | 洗錢防制 | 154 |
| Jce Humanities 45 | 指考人文科目 | 105 |
| Introduction to Law 46 | 法律概論 | 268 |
| Taxation 47 | 稅務 | 422 |
| Trust Practice 48 | 信託實務 | 451 |
| Administrative Law 49 | 行政法 | 472 |

Table 11: Overview of subjects in Humanities

| Subject Name | Chinese Name | Questions |
|---|---|---|
| Dentistry 50 | 牙醫學 | 448 |
| Traditional Chinese Medicine Clinical Medicine 51 | 中醫臨床醫學 | 314 |
| Technical Skills 52 | 技術工相關 | 452 |
| Culinary Skills 53 | 餐旅 | 330 |
| Mechanical 54 | 機械與機電概論 | 136 |
| Logic Reasoning 55 | 邏輯思維 | 160 |
| Real Estate 56 | 房地產 | 107 |
| Music 57 | 音樂科 | 314 |
| Junior Social Studies 58 | 國中會考基測社會科 | 145 |
| TVE Design 59 | 統測__設計 | 538 |
| Trade 60 | 貿易 | 563 |
| Auditing 61 | 審計學 | 616 |
| Veterinary Pharmacology 62 | 獸醫藥理學 | 605 |
| Nautical Science 63 | 航海 | 617 |
| Veterinary Pathology 64 | 獸醫病理學 | 320 |
| Accounting 65 | 會計學 | 217 |
| Fire Science 66 | 火災學 | 143 |
| Optometry 67 | 視光學 | 1027 |
| Insurance Studies 68 | 保險學 | 850 |
| Pharmacology 69 | 藥理學 | 646 |
| Management Accounting 70 | 管理會計 | 244 |
| Agriculture 71 | 農業 | 173 |
| Official Document Management 72 | 機關文書 | 252 |
| Financial Analysis 73 | 財務分析 | 429 |
| Marketing Management 74 | 行銷管理 | 108 |
| Business Management 75 | 企業管理 | 160 |
| Finance Banking 76 | 金融與法規 | 155 |

Table 12: Overview of subjects in Other

**Creation of year** Due to concern of memorization, we try to use as many questions from 2023 as shown in Figure 6. Our earliest questions can be found in as early as 1998.

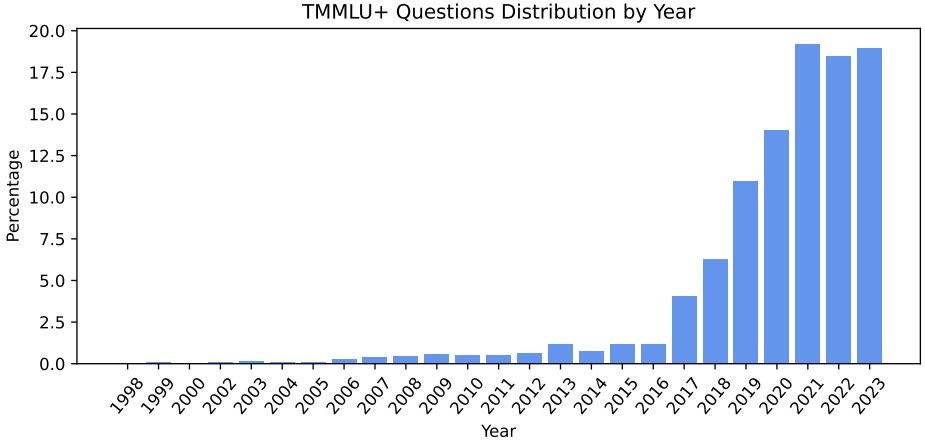

Figure 6: Year of exam questions used in TMMLU+

Compared to CMMLU and TMMLU, TMMLU+ has a slightly longer tail of the distribution in question length than the prior.

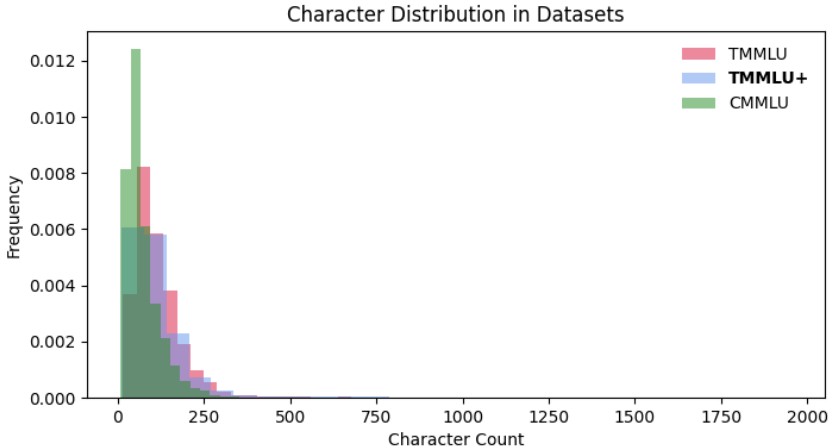

Figure 7: Character distribution comparison between TMMLU+, TMMLU and CMMLU. TMMLU+ distribution is slightly more long tail than TMMLU and CMMLU

## C  Percentage Breakdown of TMMLU+ Option Distribution

Tables 13 provide a detailed breakdown of how answer options are distributed across a wide range of subjects in TMMLU+. Each value in the table represents a percentage that corresponds to the portion of each option (A, B, C, D) within the specific subject.

| Subject | A | B | C | D |
|---|---|---|---|---|
| engineering math | 27.18 | 25.24 | 23.30 | 24.27 |
| dentistry | 25.56 | 25.56 | 24.06 | 24.81 |
| traditional chinese medicine | 25.90 | 24.10 | 24.82 | 25.18 |
| clinical psychology | 27.20 | 25.60 | 24.80 | 22.40 |
| technical | 25.12 | 24.63 | 25.62 | 24.63 |
| culinary skills | 22.60 | 26.03 | 25.34 | 26.03 |
| mechanical | 24.58 | 26.27 | 24.58 | 24.58 |
| logic reasoning | 23.02 | 25.90 | 23.74 | 27.34 |
| real estate | 23.91 | 28.26 | 22.83 | 25.00 |
| general principles of law | 27.36 | 21.70 | 25.47 | 25.47 |
| finance banking | 26.67 | 24.44 | 24.44 | 24.44 |
| anti money laundering | 20.90 | 26.12 | 25.37 | 27.61 |
| ttqav2 | 23.89 | 27.43 | 24.78 | 23.89 |
| marketing management | 25.81 | 24.73 | 23.66 | 25.81 |
| business management | 23.02 | 24.46 | 26.62 | 25.90 |
| organic chemistry | 23.85 | 26.61 | 22.02 | 27.52 |
| advance chemistry | 23.58 | 24.39 | 26.02 | 26.02 |
| physics | 23.71 | 21.65 | 27.84 | 26.80 |
| secondary physics | 27.68 | 25.89 | 24.11 | 22.32 |
| human behavior | 25.57 | 25.89 | 24.27 | 24.27 |
| national protection | 24.64 | 24.17 | 25.12 | 26.07 |
| jce humanities | 26.67 | 24.44 | 24.44 | 24.44 |
| politic science | 24.12 | 25.03 | 25.63 | 25.23 |
| agriculture | 27.15 | 25.83 | 23.84 | 23.18 |
| official document management | 25.23 | 25.23 | 24.32 | 25.23 |
| financial analysis | 24.61 | 25.13 | 24.87 | 25.39 |
| pharmacy | 25.32 | 24.55 | 25.83 | 24.30 |
| educational psychology | 26.14 | 23.30 | 27.84 | 22.73 |
| statistics and machine learning | 24.55 | 25.00 | 25.00 | 25.45 |
| management accounting | 26.05 | 25.58 | 24.65 | 23.72 |
| introduction to law | 27.00 | 23.63 | 23.21 | 26.16 |
| computer science | 26.44 | 24.14 | 25.29 | 24.14 |
| veterinary pathology | 23.67 | 26.15 | 26.15 | 24.03 |
| accounting | 25.65 | 25.13 | 24.61 | 24.61 |
| fire science | 24.19 | 26.61 | 24.19 | 25.00 |
| optometry | 25.54 | 25.33 | 25.11 | 24.02 |
| insurance studies | 25.53 | 25.53 | 24.34 | 24.61 |
| pharmacology | 25.82 | 24.96 | 25.13 | 24.09 |
| taxation | 24.00 | 27.20 | 23.47 | 25.33 |
| education (profession level) | 24.90 | 25.51 | 24.28 | 25.31 |
| economics | 25.70 | 24.94 | 24.68 | 24.68 |
| veterinary pharmacology | 25.19 | 25.00 | 25.00 | 24.81 |
| nautical science | 24.86 | 24.68 | 24.86 | 25.59 |
| psychology disorders | 24.49 | 24.31 | 25.97 | 25.23 |
| trust practice | 24.44 | 24.69 | 24.69 | 26.18 |
| geography of taiwan | 25.26 | 25.00 | 24.48 | 25.26 |
| physical education | 25.14 | 25.14 | 24.58 | 25.14 |
| auditing | 25.09 | 24.36 | 26.73 | 23.82 |
| administrative law | 25.48 | 25.24 | 24.52 | 24.76 |
| basic medical science | 24.95 | 23.90 | 25.37 | 25.79 |

| Subject | A | B | C | D |
|---|---|---|---|---|
| macroeconomics | 26.03 | 25.30 | 24.82 | 23.84 |
| trade | 25.30 | 25.10 | 25.70 | 23.90 |
| chinese language and literature | 25.13 | 25.63 | 24.62 | 24.62 |
| tve design | 23.96 | 26.67 | 23.96 | 25.42 |
| junior science exam | 26.29 | 23.47 | 25.35 | 24.88 |
| junior math exam | 27.43 | 23.43 | 25.14 | 24.00 |
| junior chinese exam | 25.71 | 24.57 | 24.00 | 25.71 |
| junior social studies | 24.60 | 26.19 | 23.02 | 26.19 |
| tve mathematics | 24.67 | 26.67 | 24.00 | 24.67 |
| tve chinese language | 26.09 | 23.81 | 25.26 | 24.84 |
| tve natural sciences | 24.53 | 25.47 | 24.53 | 25.47 |
| junior chemistry | 25.84 | 25.84 | 22.49 | 25.84 |
| music | 24.46 | 23.38 | 25.18 | 26.98 |
| education | 25.81 | 23.39 | 23.39 | 27.42 |
| three principles of the people | 25.90 | 23.74 | 25.90 | 24.46 |
| taiwanese hokkien | 25.58 | 23.26 | 26.36 | 24.81 |

Table 13: Breakdown of TMMLU+ option distribution

## D   Prompts used in evaluation process

Our prompts for zero-shot prompting can be found in Figure 8. The prompt for five shot and chain of thought prompt can be found in Figure 9 and Figure 10 respectively. The part where LLMs response is marked in red. In CoT prompting, we specifically ended the sentence in "Let's think step by step" such that the model would first reason about all four choices.



以下是關於台灣考試單選題，請選出正確的答案。
Here are the multi-choice questions about the Taiwan regional linguistic exam, please choose the correct answer.
問題：以下哪些用語來自台灣？
Question: Which of the following phrases came from Taiwan?

   A. 他「顏值」很高。
   B. 那個老先生往你車子衝過來是假裝摔倒的，你被「碰瓷」啦！
   C. 笑死我了「233333」
   D. 他po那個動態有什麼好稀奇的，不就「月經文」？

答案：D
Answer: D



Figure 8: An example of zero-shot evaluation in answer-only scenarios. The English translation is given beneath the corresponding Chinese text and is highlighted in blue.

以下是關於信託實務考試單選題，請選出正確的答案。
Here are the multi-choice questions about the Trust Practices Exam, please choose the correct answer.
問題：有關信託業辦理企業員工持股信託業務之敘述，下列何者錯誤？
Regarding the handling of corporate employee stock ownership trust business by the trust industry, which of the following statements is incorrect? Question:

A. 受託人之報酬得以信託財產充之

A. The trustee's compensation can be paid from the trust property.

B. 得享有信託利益

B. They are entitled to trust benefits.

C. 不得違反信託契約之約定

C. They must not violate the terms of the trust agreement.

D. 應由具有專門學識或經驗之人員為之

D. It should be conducted by personnel with specialized knowledge or experience.

答案：C
Answer: C
...[ Four more examples here ]...
問題：員工持股會之會員於企業員工持股信託期間，因個人特殊事由欲中途退出，原則上須經下列何者審核通過？
Question: For members of an employee stock ownership plan (ESOP), if they wish to withdraw from the corporate employee stock trust early due to personal special reasons, which of the following must approve the withdrawal in principle?

A. 公司負責人

A. The person in charge of the company

B. 受託人

B. The trustee

C. 員工持股會代表人

C. The representative of the ESOP

D. 員工持股會全體會員

D. All members of the ESOP

答案：C
Answer: C

Figure 9: An example of few-shot evaluation in answer-only scenarios. The English translation is given beneath the corresponding Chinese text and is highlighted in blue.

以下是關於貿易考試單選題，請選出正確的答案。
Here are the multi-choice questions about the trade exam, please choose the correct answer.
...[ Five chain-of-thought examples here ]...
問題：關於貿易流程，下列敘述何者正確？
Question: Regarding the trade process, which of the following statements is correct?

A. 進出口貨物皆須辦理檢驗及報關

A. Both imported and exported goods must undergo inspection and customs declaration.

B. 一般於報價後才進行信用調查

B. Credit investigation is generally conducted after quoting a price.

C. 中國輸出入銀行免費為廠商辦理國外徵信

C. The Export-Import Bank of China provides free foreign credit investigation services for manufacturers.

D. 進口報關時不一定要繳反傾銷稅

D. It is not always necessary to pay anti-dumping duties when declaring imports.

答案：讓我們一步一步思考，首先，首先，選項A並不完全正確，因為並非所有的進出口貨物都需要辦理檢驗及報關，這取決於貨物的種類和目的地國家的法規。其次，選項B的敘述也不對，一般來說，信用調查應該在報價之前進行，以確保買家有足夠的信用來完成交易。再者，選項C中的中國輸出入銀行免費為廠商辦理國外徵信，這個說法並不正確，因為徵信通常是需要費用的。因此，選項D是正確的，進口報關時不一定要繳反傾銷稅，這取決於貨物的種類和目的地國家的法規。反傾銷稅是為了保護國內產業不受到外國廉價商品的傾銷影響，但並非所有的進口商品都需要繳納。所以答案是D。

Answer: Let's think step by step. Firstly, Option A is not entirely correct, as not all import and export goods need to undergo inspection and customs declaration, which depends on the type of goods and the regulations of the destination country. Secondly, the statement in Option B is also incorrect; generally, credit investigations should be conducted before quoting to ensure that the buyer has sufficient credit to complete the transaction. Furthermore, the statement in Option C that the Export-Import Bank of China handles overseas credit information for manufacturers for free is not accurate, as credit investigation usually requires a fee. Therefore, Option D is correct; paying anti-dumping duties is not always necessary when declaring imports, depending on the type of goods and the regulations of the destination country. Anti-dumping duties are meant to protect domestic industries from the impact of cheap foreign goods being dumped, but not all imported goods are required to pay them. Therefore, the answer is D

Figure 10: An example of CoT in answer-only scenarios. The English translation is given beneath the corresponding Chinese text. The English translation appears below the corresponding Chinese text. The text highlighted in blue serves as our prompt, while the text in red was generated by GPT-4.

# E  Detailed Description of Models

This section outlines the specifics of the models utilized in our investigation.

**GPT-4 and GPT-3.5-turbo** represent OpenAI's advancements in conversational models. Initially, these models undergo supervised fine-tuning leveraging a dataset comprising user-assistant dialogues. Subsequently, reinforcement learning from human feedback (RLHF) further refines their capabilities. For this study, we employ versions of these models that were current as of June 13th, marking the latest updates available to us.

**LLaMA and LLaMA 2** epitomize Meta's efforts in enhancing large, decoder-only language models. Both iterations employ Rotary Positional Embedding (RoPE) for positional encoding and Root Mean Square Layer Normalization (RMSNorm) for model normalization. LLaMA 2 distinguishes itself by undergoing an extended pretraining phase on a diversified and more comprehensive dataset.

**Qwen** is pretrained on up to 3 trillion tokens, including Chinese, English, multilingual texts, code, and mathematics, covering general and professional fields.

**Qwen1.5-14B-Chat** incorporate SwiGLU activation, RoPE, and multi-head attention mechanisms. It offers enhanced capabilities for managing multiple languages and facilitating chat interactions, capable of supporting up to 32K tokens in context length.

**Atom-7B** is developed from the LLaMA 7B foundation, augmenting its vocabulary with an additional 32,000 Simplified Chinese tokens. This model benefits from further pretraining on a diverse and unspecified corpus, with a focus on sequences up to 4,000 tokens in length.

**DeepSeek-LLM-7B-Base** is a fresh instantiation of the LLaMA architecture, pretrained from the ground up on an extensive 2 trillion token dataset, encompassing both English and Chinese languages.

**DeepSeek-LLM-7B-Chat** evolves from the Deepseek-llm-7b-base, undergoing fine-tuning on a proprietary instructional dataset, the specifics of which remain undisclosed.

**Yayi-7b** builds upon the BLOOMZ 7B model, engaging in continuous pretraining on a selectively curated corpus of Simplified Chinese and English texts.

**Yayi-7b-llama2** mirrors the approach of Yayi-7b, albeit starting from the LLaMA 2 7B model, to undertake continuous pretraining on a similarly filtered corpus of Simplified Chinese and English texts.

**Blossom-v3.1-mistral-7B** is developed from Mistral-7B model and only further fine-tuned on Chinese instruction dataset.

**Breeze-7B-Instruct-v0.1** enhances the original Mistral-7B model by incorporating an additional 30,000 Traditional Chinese tokens into its vocabulary. This expanded model then undergoes further pretraining on a substantial, though unspecified, volume of Traditional Chinese data.

**Mistral-7b-80k** extends the Mistral 7B framework, focusing its additional pretraining efforts on Simplified Chinese datasets.

**BlueLM-7B-Base** is GPT-Neo architecture language model pre-trained on high-quality data with 2.6 trillion tokens in English and Simplified Chinese.

**Claude-3.0 Opus** developed by Anthropic, is trained using a combination of unsupervised learning and a technique called Constitutional AI. It involves pretraining the model on a massive dataset of text and image data to acquire language and vision capabilities. Subsequently, Constitutional AI guides the model's alignment with human values by explicitly specifying ethical and behavioral principles.

**Gemini-pro** is a multimodal model developed by Google AI, trained on a massive and versatile dataset to seamlessly process a combination of text, audio, and visual inputs. It is capable of generating outputs that integrate both images and text.

**Yi-6B-Chat and Yi-34B-Chat** are pre-trained on 3 trillion tokens, with the training data updated through June 2023. Both of them are trained with 4K sequence length and can be increased to 32K tokens during inference.

**Yi-9B** undergoes continuous training, building upon the Yi-6B model with 0.8T tokens.

**ChatGLM3-6B** retains the foundational structure of its predecessors, introducing an newly designed prompt format to support function calls, code interpretation, and the handling of intricate scenarios such as agent tasks.

**Taiwan-LLM-7B-v2.1-chat** is designed specifically for Traditional Chinese, emphasizing the unique linguistic and cultural aspects of Taiwan. Originating from an extensive foundational model, it incorporates a wide range of Taiwanese text sources and is enhanced through supervised fine-tuning.

**Chinese-Llama-2-7b** enhances the original Llama 2 by undergoing additional pretraining with Chinese text data around 120G and incorporating a more extensive vocabulary of size 55,296 for broadening coverage. It accommodates a 4K context size, with the capability to extend beyond 18K through the application of the NTK method.

# F   Detailed Evaluation Results for the Top 4 Performing Models

Table 14 presents a comprehensive accuracy breakdown across numerous subjects for Qwen-72B, GPT-4, Qwen-14B, and Gemini-Pro.

| Subject | GPT-4 | Qwen-14B | Qwen-72B | Gemini-pro |
|---|---|---|---|---|
| engineering math | 47.57 | 27.18 | 50.49 | 24.27 |
| dentistry | 66.67 | 50.13 | 59.65 | 52.88 |
| traditional chinese medicine | 43.53 | 51.44 | 68.71 | 43.53 |
| clinical psychology | 77.60 | 52.00 | 71.20 | 66.40 |
| technical | 66.92 | 59.70 | 66.42 | 61.97 |
| culinary skills | 60.62 | 56.51 | 66.10 | 71.43 |
| mechanical skills | 66.10 | 59.32 | 77.12 | 63.36 |
| logic reasoning | 34.53 | 32.37 | 40.29 | 35.25 |
| real estate | 43.48 | 45.65 | 48.91 | 40.18 |
| general principles of law | 61.32 | 51.89 | 67.92 | 50.94 |
| Finance Banking | 61.48 | 48.89 | 52.59 | 47.41 |
| anti money laundering | 70.90 | 73.88 | 79.85 | 61.94 |
| ttqav2 | 78.76 | 72.57 | 86.73 | 71.68 |
| marketing management | 77.42 | 76.34 | 80.65 | 70.54 |
| business management | 70.50 | 59.71 | 69.78 | 60.43 |
| organic chemistry | 66.06 | 46.79 | 66.97 | 47.93 |
| advance chemistry | 53.66 | 47.15 | 52.85 | 50.41 |
| physics | 60.82 | 43.30 | 64.95 | 44.33 |
| secondary physics | 63.39 | 52.68 | 61.61 | 42.86 |
| human behavior | 73.46 | 66.02 | 78.64 | 63.11 |
| national protection | 63.51 | 62.56 | 74.88 | 64.93 |
| jce humanities | 63.33 | 58.89 | 76.67 | 60.00 |
| politic science | 75.18 | 50.85 | 73.67 | 59.30 |
| agriculture | 47.68 | 36.42 | 56.95 | 39.07 |
| official document management | 48.65 | 50.45 | 65.77 | 40.09 |
| financial analysis | 56.28 | 37.17 | 52.88 | 37.96 |
| pharmacy | 49.62 | 34.53 | 49.87 | 38.50 |
| educational psychology | 76.14 | 60.23 | 77.27 | 56.57 |
| statistics and machine learning | 71.88 | 56.70 | 65.18 | 54.27 |
| management accounting | 47.44 | 35.35 | 48.84 | 37.22 |
| introduction to law | 57.81 | 47.26 | 62.45 | 50.63 |
| computer science | 79.89 | 66.67 | 81.03 | 62.64 |
| veterinary pathology | 63.60 | 43.11 | 63.60 | 42.05 |
| accounting | 34.55 | 29.32 | 33.51 | 29.84 |
| fire science | 33.87 | 45.16 | 49.19 | 36.29 |
| optometry | 53.26 | 40.76 | 51.30 | 36.52 |
| insurance studies | 56.97 | 47.24 | 63.03 | 43.95 |
| pharmacology | 77.82 | 50.09 | 68.28 | 53.14 |
| taxation | 40.27 | 35.20 | 41.60 | 33.87 |
| education (profession level) | 56.38 | 39.51 | 57.41 | 43.00 |
| economics | 68.45 | 40.97 | 59.29 | 44.27 |

| Subject | GPT-4 | Qwen-14B | Qwen-72B | Gemini-pro |
|---|---|---|---|---|
| veterinary pharmacology | 83.89 | 56.67 | 77.96 | 62.96 |
| nautical science | 52.45 | 38.66 | 51.18 | 41.85 |
| psychology disorders | 75.69 | 64.09 | 78.82 | 66.36 |
| trust practice | 47.38 | 43.39 | 55.61 | 40.65 |
| geography of taiwan | 69.92 | 58.20 | 73.96 | 65.10 |
| physical education | 61.45 | 41.90 | 68.72 | 45.81 |
| auditing | 56.55 | 42.18 | 61.45 | 42.55 |
| administrative law | 51.19 | 35.48 | 56.90 | 43.57 |
| basic medical science | 83.54 | 47.69 | 77.04 | 56.81 |
| macroeconomics | 63.99 | 45.50 | 62.53 | 42.86 |
| trade | 47.01 | 31.08 | 54.38 | 35.79 |
| chinese language and literature | 46.73 | 51.76 | 67.34 | 45.96 |
| tve design | 80.42 | 67.08 | 79.58 | 65.21 |
| junior science exam | 71.36 | 59.15 | 76.06 | 57.01 |
| junior math exam | 40.00 | 40.00 | 49.14 | 29.71 |
| junior chinese exam | 80.57 | 82.29 | 85.71 | 72.57 |
| junior social studies | 65.87 | 74.60 | 79.37 | 58.73 |
| tve mathematics | 33.33 | 33.33 | 36.67 | 30.00 |
| tve chinese language | 73.71 | 73.91 | 85.09 | 66.05 |
| tve natural sciences | 69.81 | 58.49 | 74.53 | 59.29 |
| junior chemistry | 54.07 | 43.54 | 49.28 | 37.32 |
| music | 58.27 | 52.52 | 67.99 | 51.58 |
| education | 66.13 | 55.65 | 65.32 | 54.03 |
| three principles of the people | 77.70 | 69.06 | 80.58 | 68.35 |
| taiwanese hokkien | 27.13 | 33.33 | 42.64 | 34.88 |

Table 14: Breakdown of model performance in the top 4 models in zero-shot learning

## G  Detailed results of various models on different benchmarks

Table 15 presents evaluation results on Simplified Chinese benchmarks and TMMLU+ to assess whether models exhibit superior performance on simplified benchmarks. *It should be noted that the performance results of GPT-4 on C-Eval and CMMLU were individually reported in Huang et al. (2023) and Li et al. (2023), respectively.

| Model | C-Eval | CMMLU | TMMLU+ |
|---|---|---|---|
| Breeze-7B-Instruct-v1.0 | 0.468 | 0.460 | 0.286 |
| Qwen-1.8B | 0.548 | 0.516 | 0.338 |
| Qwen-14B | 0.708 | 0.713 | 0.514 |
| Qwen-72B | 0.825 | 0.844 | 0.647 |
| Qwen-7B | 0.626 | 0.625 | 0.520 |
| Taiwan-LLM-13B-v2.0-chat | 0.386 | 0.401 | 0.223 |
| Taiwan-LLM-7B-v2.1-chat | 0.368 | 0.363 | 0.051 |
| GPT-4* | 0.687 | 0.710 | 0.675 |

Table 15: Five-shot results of various models on different benchmarks

## H Comparison between rule-based parser and prompt-based parser

Previous work (Kojima et al., 2022; Wei et al., 2022) that prompt-based parsers often achieve higher scores compared to rule-based parsers. While a prompt-based parser is much more versatile than a rule-based parser as prompt response format may differ significantly between LLMs, due to cost consideration, this former method is not widely adopted as the default answer extraction method. In this section, we compare the difference between these two as shown in Table 16. For the prompt-based parser, we utilize gemini-1.5-flash to extract the final answer.

| Model | STEM | | Soc. Sci. | | Humanities | | Other | | Average | |
|---|---|---|---|---|---|---|---|---|---|---|
| | LLM | String | LLM | String | LLM | String | LLM | String | LLM | String |
| GPT-4 | 62.59 | 60.4 | 70.29 | 67.4 | 59.35 | 56.0 | 61.11 | 57.6 | 63.33 | 60.3 |
| Gemini-pro | 44.29 | 45.4 | 57.92 | 57.3 | 49.85 | 48.8 | 48.50 | 48.2 | 50.14 | 49.9 |
| Yi-34B-Chat | 50.16 | 40.2 | 64.83 | 56.8 | 57.05 | 54.0 | 56.74 | 47.6 | 57.19 | 49.6 |
| GPT-3.5-turbo | 43.68 | 41.6 | 49.00 | 46.7 | 39.82 | 36.7 | 45.21 | 42.0 | 44.43 | 41.8 |
| Qwen-7B-Chat | 35.36 | 37.5 | 44.96 | 45.5 | 40.64 | 38.1 | 40.25 | 39.0 | 40.30 | 40.0 |
| Yi-9B | 28.69 | 28.7 | 44.41 | 44.6 | 48.53 | 48.5 | 36.70 | 36.8 | 39.58 | 39.6 |
| Breeze-7B-Instruct-v1.0 | 33.92 | 21.7 | 45.11 | 25.2 | 41.18 | 24.5 | 38.04 | 23.0 | 39.56 | 23.6 |
| Taiwan-LLM-13B | 27.01 | 18.5 | 32.45 | 27.6 | 27.78 | 17.8 | 28.53 | 21.5 | 28.94 | 21.3 |
| Taiwan-LLM-7B | 27.42 | 15.0 | 26.01 | 16.2 | 25.95 | 15.0 | 26.34 | 16.2 | 26.43 | 15.6 |

Table 16: Comparison of Models Across Different Fields with prompt-based parser (LLM) and rule-based parser (String) as answer extraction method

## I Data Examples for Each Subject

問題：給定兩向量$\mathbf{u} = [1\ 1\ 2]^{\mathrm{T}}$及$\mathbf{v} = [2\ -1\ 1]^{\mathrm{T}}$，下列選項何者錯誤？
Question: Given two vectors $\mathbf{u} = [1\ 1\ 2]^{\mathrm{T}}$ and $\mathbf{v} = [2\ -1\ 1]^{\mathrm{T}}$, which of the following options is incorrect?

A. 此兩向量的外積（cross product）為$[-3\ -3\ 3]^{\mathrm{T}}$
A. The cross product of these two vectors is $[-3\ -3\ 3]^{\mathrm{T}}$

B. 此兩向量的夾角為$\pi/3$
B. The angle between these two vectors is $\pi/3$

C. 此兩向量的內積（inner product）為3
C. The inner product of these two vectors is 3

D. 此兩向量的範數（norm）乘積為6
D. The product of the norms of these two vectors is 6

答案：A
Answer: A

Figure 11: An "engineering math" example. The English translation is given beneath the corresponding Chinese text and is highlighted in blue.

問題：在氯氣(Cl2)和乙烷(ethane)的UV 照光的反應中，下面哪一步反應是傳播步驟 (propagation event) ？
Question: In the UV-irradiated reaction between chlorine gas (Cl2) and ethane, which of the following steps is a propagation event?

$$I) \ Cl\cdot +CH_3\text{-}CH_3 \rightarrow CH_3\text{-}CH_3\text{-}Cl + H\cdot$$
$$II) \ Cl\cdot +CH_3\text{-}CH_3 \rightarrow CH_3\text{-}H_2C\cdot +HCl$$
$$III) \ Cl\cdot +CH_3\text{-}H_2C\cdot \rightarrow CH_3\text{-}CH_2\text{-}Cl$$
$$IV) \ Cl_2 + CH_3\text{-}H_2C\cdot \rightarrow CH_3\text{-}CH_2\text{-}Cl + Cl\cdot$$
$$V) \ Cl_2 + UV \ light \rightarrow C\,l\cdot +Cl\cdot$$

A. II 與IV

A. II and IV

B. I 與V

B. I and V

C. II, III 與IV

C. II, III and IV

D. I 與IV

D. I and IV

答案：A

Answer: A

Figure 12: An "organic chemistry" example. The English translation is given beneath the corresponding Chinese text and is highlighted in blue.

問題：於下列何種物質作業環境中較易引起癌症？
Question: Which of the following substances in the work environment are more likely to cause cancer?

A. 五氯酚

A. Pentachlorophenol

B. 甲苯。

B. Toluene

C. 二甲苯

C. Xylene

D. 一氧化碳

D. Carbon Monoxide

答案：A

Answer: A

Figure 13: An "advance chemistry" example. The English translation is given beneath the corresponding Chinese text and is highlighted in blue.

問題：有一交流發電機，其線圈面積為0.03平方公尺，線圈共20匝，以每分鐘600轉的固定速度，在0.2特斯拉的均勻磁場中旋轉，則此發電機的最大感應電動勢約為多少伏特？
Question: There is a AC generator with a coil area of 0.03 square meters, a total of 20 turns of the coil, and rotates at a fixed speed of 600 revolutions per minute in a uniform magnetic field of 0.2 Tesla. What is the approximate maximum induced electromotive force of this generator in volts?

A. 15

A. 15

B. 7.5

B. 7.5

C. 30

C. 30

D. 2.5

D. 2.5

答案：B
Answer: B

Figure 14: A "physics" example. The English translation is given beneath the corresponding Chinese text and is highlighted in blue.

問題：核反應時，有$10^{-3}$公克的質量損失可以轉換成多少熱能？
Question: In a nuclear reaction, how much heat can be transformed from a mass loss of $10^{-3}$ grams?

A. $9 \times 10^9$ 卡

A. $9 \times 10^9$ cal

B. $9 \times 10^{10}$ 卡

B. $9 \times 10^{10}$ cal

C. $2.14 \times 10^{10}$ 卡

C. $2.14 \times 10^{10}$ cal

D. $2.14 \times 10^9$ 卡

D. $2.14 \times 10^9$ cal

答案：C
Answer: C

Figure 15: A "secondary physics" example. The English translation is given beneath the corresponding Chinese text and is highlighted in blue.

問題：使用同一套毛細管黏度計測流體的黏稠度，除了需檢測流體的流動時間，尚需下列何項數據？
Question: To measure the viscosity of a fluid using the same capillary viscometer, in addition to monitoring the flow time of the fluid, which of the following data is also required?

A. 滲透壓
A. Osmotic pressure

B. 兩側液面落差
B. Differential liquid level

C. 液體密度
C. Liquid density

D. 毛細管半徑
D. Capillary radius

答案：C
Answer: C

Figure 16: A "pharmacy" example. The English translation is given beneath the corresponding Chinese text and is highlighted in blue.

問題：CIDR(Classless Inter-Domain Routing)是一種IP 地址分配方法，下列敘述何者有誤？ Question: Which of the following statements about CIDR(Classless Inter-Domain Routing), a method of IP address allocation, is incorrect?

A. CIDR 標記192.168.1.1 / 25 的子網路遮罩是255.255.255.192
A. The subnet mask of CIDR notation 192.168.1.1 / 25 is 255.255.255.192.

B. CIDR 標記172.16.0.0 / 12 是IPv4 Class B 的私有IP 範圍
B. The CIDR notation 172.16.0.0 / 12 is a private IP range of IPv4 Class B.

C. CIDR 可提高網際網路上的資料路由效率
C. CIDR can improve the routing efficiency on the Internet.

D. CIDR 可減少IP 位址浪費
D. CIDR can reduce the waste of IP addresses.

答案：A
Answer: A

Figure 17: A "computer science" example. The English translation is given beneath the corresponding Chinese text and is highlighted in blue.

問題：下列何者為刺激醛固酮分泌的因素？
Question: Which of the following factors stimulates the secretion of aldosterone?

A. 高血鈉

A. High sodium in the blood

B. 血量下降

B. Blood volume decrease

C. 高滲透度

C. High osmolarity

D. 低血鉀

D. Low potassium in blood

答案：B
Answer: B

Figure 18: A "basic medical science" example. The English translation is given beneath the corresponding Chinese text and is highlighted in blue.

問題：若針對某一社區居民隨機抽樣300 人，此樣本之平均年齡為55 歲，已知其平均年齡顯著不同於64 歲，假設社區居民的年齡呈常態分布，則下列何者比較可能是平均年齡的雙尾95%信賴區間？
Question: If a random sample of 300 residents is taken from a community, and the average age of this sample is 55 years old, and it is known that its average age is significantly different from 64 years old, and assuming that the age of the community residents is normally distributed, which of the following is more likely to be the 95% confidence interval for the average age?

A. （54, 74）

B. （45, 65）

C. （57, 71）

D. （48, 62）

答案：D
Answer: D

Figure 19: A "statistics and machine learning" example. The English translation is given beneath the corresponding Chinese text and is highlighted in blue.

問題：取一質量10 kg 材質均勻的合金，將其分成兩塊，其中一塊製成一個邊長為10 cm 的實心正立方體，另一塊製成一個質量為2 kg 的實心球，則此實心球的體積應為多少？

Question: Take a 10 kg uniform alloy, divide it into two pieces, one is made into a solid cube with a side length of 10 cm, and the other is made into a solid sphere with a mass of 2 kg. What should be the volume of this solid sphere?

A. 4000 cm$^3$

A. 4000 cm$^3$

B. 5000 cm$^3$

B. 5000 cm$^3$

C. 250 cm$^3$

C. 250 cm$^3$

D. 200 cm$^3$

D. 200 cm$^3$

答案：C

Answer: C

Figure 20: A "junior science exam" example. The English translation is given beneath the corresponding Chinese text and is highlighted in blue.

題目：珠珠家共有九人，已知今年這九人歲數的眾數、平均數、中位數、四分位距均為20，則關於3年後這九人歲數的統計量，下列敘述何者錯誤？

Question: There are nine people in Zhuzhu's family. It is known that the mode, average, median, and quartile distance of their age this year are all 20. Which of the following statements about the statistical age of these nine people three years later is incorrect?

A. 平均數是23

A. The average is 23

B. 四分位距是23

B. The quartile distance is 23

C. 中位數是23

C. The median is 23

D. 眾數是23

D. The mode is 23

答案：B

Answer: B

Figure 21: A "junior math exam" example. The English translation is given beneath the corresponding Chinese text and is highlighted in blue.

問題：人體的食物來源多樣，食物的營養成分包含有機物質及無機物質，而食物加工過程有時會加入特定食品添加物，我國政府為了替民眾的食品安全把關，訂有食品安全衛生管理法及食品安全衛生管理法施行細則，下列相關敘述何者最正確？

Question: The human body has a variety of food sources. The nutrients in food include organic and inorganic substances. Sometimes specific food additives are added during food processing. Our government has established Food Safety and Health Management Laws and implementing rules for the safety of our public's food. Which of the following statements is most correct?

A. 生酮飲食是指多攝取碳水化合物，適量攝取蛋白質及少量脂質

A. Ketogenic diet refers to consuming lot of carbohydrates, moderate protein, and a small amount of fat.

B. 營養成分中僅有醣類可以提供人類生存所需之能量

B. Only carbohydrates in nutrients can provide the energy required for human survival.

C. 食品添加物屬甜味劑、防腐劑、抗氧化劑者，應同時標示其功能性名稱

C. Food additives such as sweeteners, preservatives, and antioxidants should be labeled with their functional names.

D. 脂質屬於有機物質，可構成生物體膜系、酵素、激素、抗體及血紅素等重要成分

D. Lipids are organic substances and can be important components such as biological membranes, enzymes, hormones, antibodies, and hemoglobin.

答案：C
Answer: C

Figure 22: A "tve natural sciences" example. The English translation is given beneath the corresponding Chinese text and is highlighted in blue.

問題：放射性$Cs^{131}$ 原子的半衰期為30 年。如120 年後，剩下約3 克，則$Cs^{131}$ 最初的質量最接近

Question: The half-life of radioactive $Cs^{131}$ atoms is 30 years. If about 3 grams remain after 120 years, what is the closest initial mass of $Cs^{131}$?

A. 50 克

A. 50 grams

B. 60 克

B. 60 grams

C. 30 克

C. 30 grams

D. 40 克

D. 40 grams

答案：A
Answer: A

Figure 23: A "junior chemistry" example. The English translation is given beneath the corresponding Chinese text and is highlighted in blue.

問題：若 $\alpha, \beta$ 是方程式 $x^2 - 5x + 6 = 0$ 的兩根，則多項式 $(x - \alpha^2)(x - \beta^2)$ 為下列何者？
Question: If $\alpha, \beta$ are the roots of the equation $x^2 - 5x + 6 = 0$ , what is the polynomial $(x - \alpha^2)(x - \beta^2)$ ?
A. $x^2 - 11x + 30$
A. $x^2 - 11x + 30$
B. $x^2 - 13x + 36$
B. $x^2 - 13x + 36$
C. $x^2 + 13x + 36$
C. $x^2 + 13x + 36$
D. $x^2 + 11x + 30$
D. $x^2 + 11x + 30$
答案：B
Answer: B

Figure 24: A "tve mathematics" example. The English translation is given beneath the corresponding Chinese text and is highlighted in blue.

問題：正子掃描（PET）與功能性磁振造影（fMRI）之差別為何？
Question: What is the difference between Positron Emission Tomography (PET) and Functional Magnetic Resonance Imaging (fMRI)?
A. 前者可較清晰的辨別個體正在聽音樂或是執行數學運算時，大腦部位的活化狀態
A. The former can more clearly identify the activation status of the brain when an individual is listening to music or performing mathematical operations.
B. 前者須注射特定化學物質於受試體內，後者則不需要
B. The former requires the injection of specific chemicals into the subject, while the latter does not.
C. 前者有水平、縱切以及橫切面的三度空間解剖掃描圖，而後者只有水平面掃描圖
C. The former has three-dimensional anatomical scan images of the horizontal, sagittal, and transverse planes, while the latter only has horizontal plane scan images.
D. 前者有較清晰的圖，可評估大腦血塊所在位置
D. The former has clearer images that can evaluate the location of cerebral blood clots.
答案：B
Answer: B

Figure 25: A "clinical psychology" example. The English translation is given beneath the corresponding Chinese text and is highlighted in blue.

問題：一種由蔥花制成的煎餅，在華人社會街頭如台灣、香港、中國大陸，以及馬來西亞、新加坡等地區常見，常作為早餐或小吃食用。該食物的名稱是：

Question: A kind of pancake made from spring onions is common on the streets of Chinese societies such as Taiwan, Hong Kong, mainland China, as well as Malaysia and Singapore. It is often eaten as breakfast or snack. The name of this food is:

A. 蔥油餅

A. Scallion pancake

B. 蛋餅

B. Egg pancake

C. 水煎包

C. Shui Jian Bao

D. 臭豆腐

D. Stinky tofu

答案：A
Answer: A

Figure 26: A "TTQAv2" example. The English translation is given beneath the corresponding Chinese text and is highlighted in blue.

問題：同性戀者從認知自己的性取向到公開承認同性戀傾向的過程為「出櫃」。一般而言，出櫃需要歷經四個階段，其順序為何？

Question: What is the general sequence of the process known as "coming out" for homosexuals, from recognizing their sexual orientation to publicly acknowledging their homosexuality?

A. 自我坦承－向親友坦承－認識圈內人－出櫃

A. Self-acknowledgment - Confessing to friends and family - Knowing people inside the circle - Coming out

B. 自我坦承－認識圈內人－向親友坦承－出櫃

B. Self-acknowledgment - Knowing people inside the circle - Confessing to friends and family - Coming out

C. 認識圈內人－向親友坦承－自我坦承－出櫃

C. Knowing people inside the circle - Confessing to friends and family - Self-acknowledgment - Coming out

D. 認識圈內人－自我坦承－向親友坦承－出櫃

D. Knowing people inside the circle - Self-acknowledgment - Confessing to friends and family - Coming out

答案：B
Answer: B

Figure 27: A "human behavior" example. The English translation is given beneath the corresponding Chinese text and is highlighted in blue.

問題：指揮中心負有何項的責任
Question: What are the responsibilities of the command center?

A. 人員安撫及心理諮商

A. Personnel appeasement and psychological counseling

B. 通報主管機關已疏散人數、收容地點及災情

B. Report to the competent authority the number of people evacuated, the place of refuge, and the disaster

C. 協助至學校避難民眾應急所需

C. Assist the people who take refuge in schools to meet emergency needs

D. 依情況調動各組織間相互支援

D. Adjust mutual support between organizations according to the situation

答案：D
Answer: D

Figure 28: A "national protection" example. The English translation is given beneath the corresponding Chinese text and is highlighted in blue.

問題：下列何項不是政治行為研究途徑的重要主張？
Question: Which of the following is not an important claim of the research approach to political behavior?

A. 非科際整合研究

A. Non-interdisciplinary integrated research

B. 以政治角色行為作為研究的基本資料

B. Using political role behavior as the basic data for research

C. 學術研究必須價值中立

C. Academic research must be value-neutral

D. 採用自然科學的方法

D. Using the methods of natural science

答案：A
Answer: A

Figure 29: A "politic science" example. The English translation is given beneath the corresponding Chinese text and is highlighted in blue.

問題：小芳買了火車票坐在指定位置上，卻遇到一位長者要求讓座。小明見狀說：「買了票就是有使用權，沒有人有任何理由要求你讓座」。以柯柏格(L. Kohlberg)的道德推理階段論來解釋，小明屬於哪一種取向？
Question: Xiao Fang bought a train ticket and sat in the designated position, but encountered an elder asking for a seat. Xiao Ming said: "If you bought a ticket, you have the right to use it. No one has any reason to ask you to give up your seat." Explained by Kohlberg's stage theory of moral reasoning, which orientation does Xiao Ming belong to?

A. 社會契約取向

A. Social contract orientation

B. 法律秩序取向

B. Legal order orientation

C. 普遍倫理取向

C. Universal ethics orientation

D. 相對功利取向

D. Relative utilitarian orientation

答案：B
Answer: B

Figure 30: An "educational psychology" example. The English translation is given beneath the corresponding Chinese text and is highlighted in blue.

問題：某教材於單元結束前，提供幾個與單元概念相關的探究主題，學生能依自己的興趣與能力進行探究，這種提供探究主題讓學生決定探究方法及結果的探究形式稱為？
Question: In a certain textbook, before the end of the unit, a few exploration topics related to the unit concept are provided. Students can explore according to their own interests and abilities. What is the form of exploration that provides exploration topics and allows students to determine the methods and results of exploration?

A. 結構性的探究

A. Structured exploration

B. 引導性的探究

B. Guided exploration

C. 開放式探究

C. Open-ended exploration

D. 食譜式探究

D. Recipe-based exploration

答案：B
Answer: B

Figure 31: An "education (profession level)" example. The English translation is given beneath the corresponding Chinese text and is highlighted in blue.

問題：寡占市場結構與其他三種類型市場結構之最大差異在於下列何者？
Question: What is the biggest difference between oligopolistic market structure and the other three types of market structure from the following?

A. 廠商做決策時會考慮到競爭者是如何做決策
A. When making decisions, firms will consider how competitors make decisions.

B. 個別廠商是價格的決定者
B. Each firm is a price determinant.

C. 不同廠商所生產之產品是異質的
C. Different firms produce heterogeneous products.

D. 不同廠商所生產之產品是同質的
D. Different firms produce homogeneous products.

答案：A
Answer: A

Figure 32: An "economics" example. The English translation is given beneath the corresponding Chinese text and is highlighted in blue.

問題：依據Tenhula & Bellack（2008）的社交技巧模式（Social Skills Model），下列何項描述屬於社交認知（social cognition）功能的範疇？
Question: According to the Social Skills Model by Tenhula & Bellack (2008), which of the following descriptions falls under the category of social cognition?

A. 正確偵測情感線索（如：面部與聲音表情）
A. Correctly detecting emotional clues (e.g., facial and voice expressions)

B. 有能力傾聽及確知對語言訊息的了解
B. The ability to listen and understand language messages

C. 從互動情境了解、有效分析社交刺激
C. Understand and effectively analyse social stimuli from interactive situations

D. 有能力在說話時使用合適的非語言行為
D. The ability to use appropriate non-verbal behaviour when speaking

答案：C
Answer: C

Figure 33: An "occupational therapy for psychological disorders" example. The English translation is given beneath the corresponding Chinese text and is highlighted in blue.

問題：台灣海峽的澎湖水道有「黑水溝」之稱，最有可能的原因為何？
Question: What's the most likely reason for the "Black Ditch" name of the Penghu Strait in the Taiwan Strait?

A. 水流方向多變使泥沙混濁

A. The variable water flow direction makes the silt muddy

B. 沉積物少使光線直達深處

B. Few sediments allow light to reach deep

C. 海峽兩側提供豐富懸浮物

C. Both sides of the strait provide abundant suspended particles

D. 海水中多為深色沉積物

D. Most of the seawater contains deep-colored sediments

答案：B
Answer: B

Figure 34: A "geography of taiwan" example. The English translation is given beneath the corresponding Chinese text and is highlighted in blue.

問題：田徑100 公尺比賽，在起跑後30～70 公尺階段，應採取何種方式來提高速度？
Question: In a 100-meter race, what should one do to increase speed during the 30 to 70-meter stage after start?

A. 同時增大步幅、步頻

A. Increase both stride length and frequency

B. 保持步頻、增加步幅

B. Keep the frequency constant, increase stride length

C. 減低步幅、增加步頻

C. Reduce stride length, increase frequency

D. 保持步幅、增加步頻

D. Keep the stride length constant, increase frequency

答案：D
Answer: D

Figure 35: A "physical education" example. The English translation is given beneath the corresponding Chinese text and is highlighted in blue.

問題：總合需求曲線為負斜率的理由，下列何者不是？
Question: Among the reasons why the aggregate demand curve is negative sloping, which one is not?

A. 替代效果

A. Substitution Effect

B. 利率效果

B. Interest Rate Effect

C. 實質餘額效果

C. Real Balance Effect

D. 理性預期效果

D. Rational Expectations Effect

答案：D
Answer: D

Figure 36: A "macroeconomics" example. The English translation is given beneath the corresponding Chinese text and is highlighted in blue.

問題：藍鼎元《東征集》記錄了非常多臺灣古地名或別稱，下面篇章與地點組合，何者有誤？
Question: Which of the following chapter and location combinations from Lan Dingyuan's "Eastern Expedition Collection", which recorded many ancient Taiwanese place names or aliases, is incorrect?

A. ⟨紀荷包嶼⟩：嘉義朴子

A. ⟨Record of Hebaoyu⟩: Jiayi Puzi

B. ⟨紀水沙連⟩：日月潭

B. ⟨Record of Shuishalian⟩: Sun Moon Lake

C. ⟨紀火山⟩：大屯火山群

C. ⟨Record of Volcano⟩: Datun Volcano Group

D. ⟨紀竹塹埔⟩：新竹

D. ⟨Record of Zhuxianpu⟩: Hsinchu

答案：C
Answer: C

Figure 37: A "chinese language and literature" example. The English translation is given beneath the corresponding Chinese text and is highlighted in blue.

問題：陸羽《茶經》論水云：「山水上，石泉又上，江水次而井水下。」
又「江水取去人遠者，井取汲多者。」其説止於此，未嘗品第天下之水味。
張又新《煎茶水記》云劉伯芻謂水有七等，以揚子江為第一，惠山石泉為第二，虎丘井第三，淮水居末。
又載羽為李季卿論水次第有二十種，江水居山水上，井水居江水上。二説皆與《茶經》不合。
水味有美惡而已，欲求天下之水一一而次第之者，謬説也。
羽之論水，惡渟浸 1 而喜泉源，故井取多汲者。江雖長，然衆水雜聚，故次山水。惟此説近物理云。
——改寫自歐陽脩〈大明水記〉 根據本文，下列敘述何者最恰當？

Question: Lu Yu states in "The Tea Classic" that "mountain water is on top, followed by stone spring, with river water next and well water at the bottom." Also "river water is drawn from afar, and well water is drawn more. Hence, it ends here, he never tasted all the waters in the world. In Zhang Youxin's "Record of Boiling Tea Water", it says Liu Bochao classified water into seven categories, ranking the Yangtze River as the first, Huishan Stone Spring as the second, Tiger Hill Well as the third, and Huai River as the last. Also, it says, Lu Yu discussed with Li Jiqing that there were twenty types of water, with river water towering over mountain water, and well water standing above river water. These two theories contradict The Tea Classic. The taste of water is just good or bad, and it is absurd to rank all the waters in the world one by one. Lu Yu's discussion of water dislikes cloudy and stagnant water and prefers spring sources, so wells are drawn more. Though the river is long, it is a gathering of multiple waters, hence it is next to mountain water. This view is closer to physical reason. –Rewritten from Ouyang Xiu's "Record of Daming Water" According to the text, which of the following statements is most suitable?

A. 張又新將天下的水按水質分成二十種

A. Zhang Youxin categorizes all the waters in the world into twenty kinds according to water quality.

B. 歐陽脩按水源的遠近和高低評定水質

B. Ouyang Xiu evaluates the water quality based on the distance and height of the water source.

C. 依《茶經》論水的標準，揚子江優於虎丘井

C. According to the standards of "The Tea Classic", Yangtze River is better than Tiger Hill Well.

D. 在《煎茶水記》中，張又新反駁劉伯芻看法

D. In "Record of Boiling Tea Water", Zhang Youxin refutes Liu Bochao's view.

答案：C
Answer: C

Figure 38: A "junior chinese exam" example. The English translation is given beneath the corresponding Chinese text and is highlighted in blue.

問題：閱讀，最主要的功用在幫助讀者明白一件事實的由來與意義，而不僅只是讓讀者懂得一件事實的知識。閱讀是一種學習，因為沒有教師在身旁指點你，所以學習獲益全靠讀者的分析與演繹。換言之，閱讀的主體是你，一切操之在你，所以其內涵就帶有積極性。再者，因是為了自己而讀，所以也需要有自律性。阿德勒和范達倫在他們的《如何閱讀一本書》裏把閱讀譬喻為棒球，讀者閱讀一本書應以捕手的態度相對。要想打贏球賽，投捕雙方的默契是不可少的基礎。書的著者就有如投手，想要捕捉到投手投來的球，最重要的是，捕手要積極地弄清楚投手的技巧與作風。唯有這樣，做捕手的我們才不會錯失飛來的球，也才能達到閱讀的目的。________，就很像這種投捕之間的關係。閱讀可以加強觀察力，觀察力會使生活更具意義。閱讀也會增加記憶力，因為記憶力是腦細胞的多方聯繫結果。閱讀也會藉別人的經驗與聯想力來加強你的聯想力，而聯想力是很多知識發現的基礎。甚至，閱讀也會增強你的安全性，因為知道別人的經驗，自己的機警性也會跟著提高。閱讀也會增加你的分析力，因為分析是閱讀不可或缺的一環。這些能力，會讓你的判斷力因而成熟，進而犀利，這會嘉惠你自己一輩子，也能惠及別人，並進而貢獻於社會。( 改寫自黃崑巖(閱讀是終身學習的唯一途徑)) 依上文，畫線處最適合填入下列哪一個選項？

Question: The main function of reading is to help readers understand the origin and meaning of a fact, not just to make readers know a fact. Reading is learning because there is no teacher around you to point you out, so the learning benefit depends entirely on the reader's analysis and interpretation. In other words, the subject of reading is you, and everything is up to you, so its connotation is proactive. Moreover, because it is for oneself, it also needs to have self-discipline. Adler and Van Doren in their "How to Read a Book" liken reading to baseball, readers should read a book with the attitude of the catcher. To win a game, the tacit understanding between the pitcher and the catcher is an indispensable basis. The author of the book is like a pitcher, the most important thing to catch the ball thrown by the pitcher is that the catcher should actively understand the pitcher's skills and style. Only in this way, we as a catcher will not miss the flying ball and can achieve the purpose of reading. ______, is very similar to this pitching relationship. Reading can enhance observation, and observation will make life more meaningful. Reading will also increase memory because memory is the result of multiple connections of brain cells. Reading will also increase your imagination through other people's experiences and imagination, and imagination is the basis of many knowledge discoveries. Even, reading will enhance your safety because knowing other people's experiences will also improve your alertness. Reading will also increase your analytical power because analysis is an indispensable part of reading. These abilities will mature your judgment, then sharpen, this will benefit you all your life, can also benefit others, and contribute to society. (Rewritten from Huang Kunkyin's "Reading is the Only Way to Lifelong Learning") According to the above, which of the following options is the most suitable for underlined?

A. 創作的靈感與能力

A. Insights and abilities to create

B. 教師與學生的互動

B. Interactions between teachers and students

C. 進攻的時機與技巧

C. Opportunity and skill to attack

D. 著者與讀者的關係

D. Relationship between the author and reader.

答案：D
Answer: D

Figure 39: A "tve chinese language" example. The English translation is given beneath the corresponding Chinese text and is highlighted in blue.

問題：「天生我材必有用」和「行行出狀元」的説法和下列哪種智力理論的觀點最相符合：
Question: Which of the following theories of intelligence is most consistent with the saying "Everyone is born with a talent that must be useful" and "Everyone can be the top scholar"?

A. 智力二因論

A. Two-Factor Theory of Intelligence

B. 智力三元論

B. Triarchic Theory of Intelligence

C. 智力多元論

C. Multiple Intelligences Theory

D. 智力結構論

D. Structural Theory of Intelligence

答案：C
Answer: C

Figure 40: An "education" example. The English translation is given beneath the corresponding Chinese text and is highlighted in blue.

問題：人類為追求經濟的發展，不斷地向大自然予取予求，近年來大自然的反撲已造成許多的災難，嚴重危害到人民的生命財産，這是因為與何種理念漸行漸遠所導致的結果？
Question: Due to mankind's pursuit of economic development, the incessant exploitation has led to nature's backlash, causing numerous disasters, and seriously endangering people's lives and property. What concept is this due to gradually straying from?

A. 産業升級

A. Industrial upgrading

B. 社會福利

B. Social welfare

C. 國營事業民營化。

C. Privatization of state-owned enterprises

D. 永續發展

D. Sustainable development

答案：D
Answer: D

Figure 41: A "three principles of the people" example. The English translation is given beneath the corresponding Chinese text and is highlighted in blue.

問題：「未曾＿＿＿＿＿先學飛，未曾＿＿＿＿＿想挽瓜。」這句諺語的空格仔內愛用啥物？
Question: Fill the blanks in the proverb "Never ＿＿＿＿＿ before learning to fly, never ＿＿＿＿＿ want to pull the melon." What should be used in the blanks?

A. 學行，掖種

A. learning to walk, arm planting

B. 行路，沃肥

B. walking, watering the fertilizer

C. 學爬，犁田

C. learning to crawl, plowing the field

D. 行路，種瓜

D. walking, planting melons

答案：A
Answer: A

Figure 42: A "taiwanese hokkien" example. The English translation is given beneath the corresponding Chinese text and is highlighted in blue.

問題：甲向乙約定購買機車一台，並交付定金新臺幣(下同)1 萬元於乙。嗣後，因可歸責於乙之事由，致該買賣契約不能履行時，依民法規定，甲得請求乙返還多少金額?
Question: How much money can A ask B to return according to the Civil Law when the contract of sale cannot be performed due to B's fault after A promised to buy a motorcycle from B and paid B a deposit of NTD 10,000?

A. 1 萬元

A. 10,000 NTD

B. 3 萬元

B. 30,000 NTD

C. 2 萬元

C. 20,000 NTD

D. 5 千元

D. 5,000 NTD

答案：C
Answer: C

Figure 43: A "general principles of law" example. The English translation is given beneath the corresponding Chinese text and is highlighted in blue.

問題：有關資恐防制，下列敘述何者錯誤？
Question: Which of the following statements about counter-terrorism financing is incorrect?

A. 我國資恐防制法指定之制裁名單，以個人、法人或團體在中華民國領域內者為限

A. The sanction list designated by our country's counter-terrorism financing law is limited to individuals, corporations, or groups within the territory of the Republic of China.

B. 我國資恐防制法之主管機關為法務部

B. The supervisory authority of our country's counter-terrorism financing law is the Ministry of Justice.

C. 資恐防制的目的是為了防制與遏止恐怖活動、組織、分子之資助行為

C. The purpose of counter-terrorism financing is to prevent and curb the financing of terrorist activities, organizations, and individuals.

D. 防制洗錢金融行動工作組織(FATF)自2001 年美國911 事件後，已將打擊資恐列為優先項目

D. The Financial Action Task Force (FATF) has made combating financing of terrorism a priority since the US 911 incident in 2001.

答案：A
Answer: A

Figure 44: An "anti money laundering" example. The English translation is given beneath the corresponding Chinese text and is highlighted in blue.

問題：依據康培寫給王晉輝的書信，敘述最適當的是：
Question: According to the letter Kang Pei wrote to Wang Jinhui, which of the following descriptions is most appropriate?

A. 康培寫這封信的目的，是拜託王晉輝代為請假

A. The purpose of Kang Pei's letter is to ask Wang Jinhui to take leave on his behalf.

B. 康培表示因延誤就醫，使病況加重且疹增多

B. Kang Pei indicated that the delay in seeking medical advice led to the worsening of the condition and increase in caruncle.

C. 由全文可推知，二人為年齡相近的長官與下屬

C. It can be inferred from the whole text that the two are bosses and subordinates of similar ages.

D. 對王晉輝的信末結尾問候語，也可用「福安」

D. The ending greeting to Wang Jinhui can also be "fu an".

答案：C
Answer: C

Figure 45: A "jce humanities" example. The English translation is given beneath the corresponding Chinese text and is highlighted in blue.

問題：依司法院大法官解釋意旨，關於化妝品廣告之事前審查，下列敘述何者錯誤？
Question: According to the interpretation of the Grand Justices of the Judicial Yuan, which of the following statements is incorrect regarding the pre-review of cosmetics advertisements?

A. 必須係為防免人民生命、身體、健康遭受直接、立即及難以回復危害之特別重要之公共利益目的
A. It must be for the especially important public interest purpose of preventing people's lives, bodies, and health from direct, immediate, and irreversible harm.

B. 係對言論自由之重大干預，原則上應為違憲
B. It is a significant intervention in freedom of speech, and should in principle be unconstitutional.

C. 其與目的之達成應有實質關聯
C. The achievement of its goal should have a substantial connection.

D. 須賦予人民獲立即司法救濟之機會
D. It should provide people with the opportunity for immediate judicial relief.

答案：C
Answer: C

Figure 46: An "introduction to law" example. The English translation is given beneath the corresponding Chinese text and is highlighted in blue.

問題：綜合所得稅中納稅義務人可享有之各種扣除額，按扣除金額由高至低進行排序，下列何選項之排序正確？1. 有配偶者之標準扣除額2. 子女教育學費特別扣除3. 身心障礙特別扣除4. 長期照顧特別扣除
Question: Which of the following options is the correct order for the various deductions that taxpayers can enjoy in the consolidated income tax, arranged from high to low by deductible amount? 1. Standard deduction for those with spouses 2. Special deduction for children's education expenses 3. Special deduction for physical and mental disability 4. Special deduction for long-term care

A. 1342
A. 1342

B. 3142
B. 3142

C. 3412
C. 3412

D. 1432
D. 1432

答案：A
Answer: A

Figure 47: A "taxation" example. The English translation is given beneath the corresponding Chinese text and is highlighted in blue.

問題：有關財産贈與稅之價值認定，下列敘述何者錯誤？
Question: Which of the following statements about the valuation of property gift tax is incorrect?

A. 外幣存款以贈與日之臺銀買匯匯率換算新臺幣計算

A. Foreign currency deposits are converted into New Taiwan dollars using the buying exchange rate of the Taiwanese Bank on the day of the gift.

B. 基金以贈與日基金淨值計算

B. Funds are calculated based on the net value of the fund on the day of the gift.

C. 上市櫃及興櫃股票以贈與日收盤價計算

C. Listed and OTC stocks are calculated based on the closing price on the day of the gift.

D. 非上市櫃股票以股票淨值計算

D. Unlisted stocks are calculated based on the net value of the stock.

答案：C
Answer: C

Figure 48: A "trust practice" example. The English translation is given beneath the corresponding Chinese text and is highlighted in blue.

問題：下列有關法規適用原則之敘述何者錯誤？
Question: Which of the following statements about the principle of legal application is incorrect?

A. 法律優於法規命令

A. The law supersedes regulatory orders

B. 法律優於緊急命令

B. The law is superior to emergency orders

C. 後法優於前法

C. The later law is superior to the former law

D. 特別法優於普通法

D. Special law is superior to ordinary law

答案：B
Answer: B

Figure 49: An "administrative law" example. The English translation is given beneath the corresponding Chinese text and is highlighted in blue.

問題：下列牙齒何者不易由牙冠來區別左側或右側？
Question: Which of the following teeth cannot be easily distinguished from the left or right by the crown?

A. 上顎乳犬齒

A. Maxillary deciduous canine

B. 上顎正中乳門齒

B. Maxillary central deciduous incisors

C. 下顎乳犬齒

C. Mandibular deciduous canine

D. 下顎正中乳門齒

D. Mandibular central deciduous incisors

答案：D
Answer: D

Figure 50: A "dentistry" example. The English translation is given beneath the corresponding Chinese text and is highlighted in blue.

問題：依《針灸科學》有關遺精之敘述，下列何者錯誤？
Question: According to "Acupuncture Science", which of the following statements about seminal emission is wrong?

A. 先針三陰交、陰陵泉二穴

A. First, needle the Sanyinjiao and Yinlingquan

B. 睡夢中射精名為滑精

B. Ejaculation in sleep is named as spermatorrhea

C. 繼針氣海、關元、中極、腎俞、志室，並灸腎俞、關元、志室各七八壯

C. Continue to needle Qihai, Guanyuan, Zhongji, Shenshu, Zhishi, and moxibustion at Shenshu, Guanyuan and Zhishi

D. 遺精的病因為神經衰弱、精神疲勞、飲酒過度、鄰近臟器發炎，又有因手淫過甚、慾念妄動及縱慾傷損而起

D. The cause of seminal emission can be nervous weakness, mental fatigue, excessive drinking, inflammation of nearby organs, and it can also be caused by excessive masturbation, lustful thoughts and indulgence

答案：B
Answer: B

Figure 51: A "traditional chinese medicine clinical medicine" example. The English translation is given beneath the corresponding Chinese text and is highlighted in blue.

問題：配電場所用戶預埋管之處理方法為
Question: What is the treatment method for the pre-embedded pipe of the power distribution place user?

A. 不予處理

A. Do not dealing with it

B. 施做單側防水。

B. Implement unilateral waterproofing

C. 僅做埋管即可

C. Just do the embedding pipe is enough

D. 預埋管兩端應加密封防水

D. The ends of the embedding pipe should be sealed for waterproofing

答案：D
Answer: D

Figure 52: A "technical" example. The English translation is given beneath the corresponding Chinese text and is highlighted in blue.

問題：食材驗收時應注意之事項，下列敘述何者正確
Question: Which of the following is the correct statement regarding things that should be paid attention to when receiving ingredients?

A. 運輸條件無須驗收

A. There is no need to check the transport conditions.

B. 冷凍食品包裝上有水漬/冰晶時，不宜驗收。

B. Frozen food packages should not be accepted if there are water stains or ice crystals on them.

C. 現場合格者驗收，無須記錄

C. Those who pass on-site should be received, no record is necessary.

D. 採購及驗收應同一人辦理

D. The same person should handle both purchasing and receiving.

答案：B
Answer: B

Figure 53: A "culinary skills" example. The English translation is given beneath the corresponding Chinese text and is highlighted in blue.

問題：防治蟲害最好的方法是
Question: What is the best method to control insect pests?

A. 拍打
A. Patting

B. 使用殺蟲劑
B. Using insecticides

C. 網子捕捉
C. Using a net to catch

D. 清除孳生源。
D. Clearing the source of infestation

答案：D
Answer: D

Figure 54: A "mechanical" example. The English translation is given beneath the corresponding Chinese text and is highlighted in blue.

問題：句中字未依順序排列，且有多餘的字，請找出多餘的字：不夠投半話句機多
Question: The words in the sentence are not arranged in order, and there are extra words, please find out the extra words: 不夠投半話句機多

A. 話
B. 句
C. 夠
D. 機
答案：C
Answer: C

Figure 55: A "logic reasoning" example. The English translation is given beneath the corresponding Chinese text and is highlighted in blue.

問題：某三層樓獨棟透天新成屋，建物登記面積為40 坪，坐落基地登記面積為20 坪，房地之正常價格為1200 萬元，經參酌當地市場調查資料，運用估價方法計算出建物價值比率為占房地價格的30%，請問該基地單價應為多少？

Question: For a three-story detached new building, the registered area of the building is 40 pings, the registered area of the base is 20 pings, if the normal price of the house and land is 12 million yuan. Considering the local market survey data, the building value ratio calculated using the valuation method accounts for 30% of the house price. Please, find the unit price of this base.

A. 42 萬元/坪

A. 420,000 NTD/sqft

B. 60 萬元/坪

B. 600,000 NTD/sqft

C. 18 萬元/坪

C. 180,000 NTD/sqft

D. 30 萬元/坪

D. 300,000 NTD/sqft

答案：A
Answer: A

Figure 56: A "real estate" example. The English translation is given beneath the corresponding Chinese text and is highlighted in blue.

問題：有關「車鼓調」的敘述何者為非？
Question: Which of the following statements about "Chegu Diao" is false?

A. 「鼓」指的是鈴鼓

A. "Drum" refers to the bell drum.

B. 為京劇中的一種曲調

B. It's a type of tune in Peking Opera.

C. 歌曲《桃花過渡》就是此種樂曲形式的代表之一

C. The song "Peach Blossoms Across the River" is one of the representatives of this musical form.

D. 「車」指的是成對的長竹塊

D. "Che" refers to a pair of long bamboo pieces.

答案：B
Answer: B

Figure 57: A "music" example. The English translation is given beneath the corresponding Chinese text and is highlighted in blue.

問題：蘭姆酒是古巴的名產之一，因味道濃烈而受到歡迎。蘭姆酒為當地發展熱帶栽培業時，使用某種經濟作物的副產品製作而成。這種經濟作物也曾經普遍種植於臺灣南部平原地帶。回溯這種經濟作物在古巴的發展，與西班牙的殖民有關。當時西班牙為了栽培經濟作物與挖掘礦產，需要大量勞力，但原住民因感染外來傳染病而大量死亡，遂引入非洲黑人填補勞動力的空缺。今日古巴的人口結構，以白人為主，具有黑人血統者，約占全國人口比例三分之一，而原住民已經相當少見。二十世紀時，有人將蘭姆酒和源於美國的可樂調製成「自由古巴」，如圖(十三)所示。此調酒的命名有個歷史性的傳說，象徵美國支持十九世紀古巴獨立運動。當時美國政壇對古巴獨立運動曾做出評論：「關於任何歐洲國家對美洲的入侵，美國有權進行干預，使歐洲勢力撤離。西班牙政府正失去對古巴的控制，且無力保護當地美國公民的生命和財產，因此美國政府應介入使當地恢復和平，並確保古巴獲得獨立。」62a69d5962675.jpg 關於文中所提到臺南市的例子，下列何項敘述最適當？

Question: Rum is one of Cuba's famous products and is popular because of its strong flavor. Rum was made from a by-product of a certain cash crop when the local area developed tropical cultivation. This cash crop was also widely planted in the southern plains of Taiwan. Tracing back the development of this economic crop in Cuba, it is related to Spanish colonization. At that time, Spain needed a lot of labor to cultivate economic crops and mine minerals, but the aboriginals died in large numbers due to infectious diseases from abroad, so Africans were introduced to fill the labor gap. Today, the population of Cuba is predominantly white, with those of black ancestry accounting for about a third of the national population, while aborigines are rare. In the twentieth century, someone mixed rum with cola originated from the United States to make "Free Cuba", as shown in Figure 13. The name of this cocktail has a historical legend, symbolizing the United States' support for Cuba's independence movement in the 19th century. At the time, the US political arena made a comment on Cuba's independence movement: "Regarding any European country's invasion of the Americas, the United States has the right to intervene and withdraw European forces. The Spanish government is losing control of Cuba and is unable to protect the lives and property of American citizens there, so the US government should intervene to restore peace and ensure Cuba's independence." 62a69d5962675.jpg Regarding the example mentioned in Tainan City, which of the following statements is most appropriate?

A. 由立法院制定法律對違規者處以行政處分

A. The legislature enacts laws to administer administrative penalties on violators.

B. 由地方立法機關訂定規定對違規者處以刑事處罰

B. Local legislative bodies establish regulations to impose criminal penalties on violators.

C. 由地方立法機關訂定規定對違規者處以行政處分

C. Local legislative bodies establish regulations to administer administrative penalties on violators.

D. 由立法院制定法律對違規者處以刑事處罰

D. The legislature enacts laws to impose criminal penalties on violators.

答案：C

Answer: C

Figure 58: A "junior social studies" example. The English translation is given beneath the corresponding Chinese text and is highlighted in blue.

問題：有關美學( aesthetics )的敘述，下列何者正確？
Question: Which of the following statements about aesthetics is correct?

A. 康德( Immanuel Kant )認為只有上帝，而不是人類，才能扮演解釋藝術理論與美感的角色的

A. Immanuel Kant believed that only God, not humans, can play a role in explaining art theory and aesthetics.

B. 休謨( David Hume )用品味( taste )指稱人類感受和判斷某件藝術品的美感能力

B. David Hume used "taste" to refer to the human ability to feel and judge the aesthetics of an artwork.

C. 美學一詞源自希臘字「aesthesis」，原意指人類的邏輯推理能力

C. The term "aesthetics" comes from the Greek word "aesthesis", originally referring to human's ability of logical reasoning.

D. 亞里斯多德( Aristotle )最早用美學一詞指稱研究美感經驗的學問

D. Aristotle was the first to use the term "aesthetics" to refer to the study of aesthetic experience.

答案：B
Answer: B

Figure 59: A "tve design" example. The English translation is given beneath the corresponding Chinese text and is highlighted in blue.

問題：關於forfaiting，下列敘述何者正確？
Question: Which of the following statements about forfaiting is correct?

A. 相關信用狀須經保兌

A. The related letter of credit must be confirmed.

B. 相關匯票須經開狀銀行承兌

B. The related bill must be accepted by the issuing bank.

C. 相關票據到期如不獲兌付，出口商應無條件返還票款

C. If the relevant note is not honored at maturity, the exporter should unconditionally return the ticket money.

D. 最適合以買方遠期信用狀項下匯票辦理

D. The bill under the buyer's term letter of credit is the most suitable for handling this.

答案：B.
Answer: B.

Figure 60: A "trade" example. The English translation is given beneath the corresponding Chinese text and is highlighted in blue.

問題：下列有關分析性程序的敘述，何者錯誤？
Question: Which of the following statements about analytic procedures is incorrect?

A. 較適用於量大且變動可推估之交易
A. More suitable for large and predictable transactions

B. 得協助查核人員作成整體結論
B. Can assist auditors in making overall conclusions

C. 較適用於資料間關係將持續存在的情況
C. More suitable for cases where the relationship between data will continue to exist

D. 必須與細項測試結合，始得對個別聲明執行證實程序
D. Must be combined with detailed tests to verify individual statements

答案：D
Answer: D

Figure 61: An "auditing" example. The English translation is given beneath the corresponding Chinese text and is highlighted in blue.

問題：有關分布體積，下列敘述何者錯誤？
Question: Which of the following statements about distribution volume is wrong?

A. 注射用麻醉劑再分布現象會延長麻醉作用
A. The redistribution phenomenon of injectable anesthetics will prolong the anesthetic effect

B. 血漿蛋白與藥物結合程度為影響分布體積因子之一
B. The degree of plasma protein binding with drugs is one of the factors affecting the distribution volume

C. 分布體積決定初始劑量（loading dose）
C. The distribution volume determines the initial dose (loading dose)

D. 是體內藥物總量與血漿藥物濃度的比值.
D. It is the ratio of the total amount of drugs in the body to the concentration of drugs in plasma.

答案：A
Answer: A

Figure 62: A "veterinary pharmacology" example. The English translation is given beneath the corresponding Chinese text and is highlighted in blue.

問題：檢查象限軟鐵球是否含半永久磁性，應將該軟鐵球移至磁羅經近處，下列那一項檢查結果可表示該軟鐵球已留存磁性？
Question: To check if a quadrant soft iron ball has semi-permanent magnetism, this soft iron ball should be moved to the vicinity of a magnetic compass. Which of the following inspection results can indicate that this soft iron ball has retained magnetism?

A. 作45°轉動，磁針偏移2°以上

A. Rotate 45°, the compass needle deviates more than 2°

B. 作90°轉動，磁針偏移1°以上

B. Rotate 90°, the compass needle deviates more than 1°

C. 作45°轉動，磁針偏移1°以上

C. Rotate 45°, the compass needle deviates more than 1°

D. 作90°轉動，磁針偏移2°以上

D. Rotate 90°, the compass needle deviates more than 2°

答案：D
Answer: D

Figure 63: A "nautical science" example. The English translation is given beneath the corresponding Chinese text and is highlighted in blue.

問題：下列何者與牛心內膜礦物質化的發生無關？
Question: Which of the following is irrelevant to the mineralization of the endocardium in bovine hearts?

A. 攝食過量維生素D

A. Overeating of vitamin D

B. 酮病

B. Ketosis

C. 採食過量含鈣植物所引發之中毒

C. Poisoning caused by overeating calcium-rich plants

D. 尿毒症

D. Uremia

答案：B
Answer: B

Figure 64: A "veterinary pathology" example. The English translation is given beneath the corresponding Chinese text and is highlighted in blue.

問題：X1 年初甲公司以$3,000,000 買入具污染性設備，另支付$200,000 安裝費，設備估計耐用3 年，甲公司預估3 年後報廢需花費$500,000 拆卸處理費，該公司資本率為10%，報廢該設備之處理無任何相關法規規定，則該設備成本應為：

Question: At the beginning of year X1, company A bought a polluting device for $3,000,000, and paid another $200,000 for installation. The device is expected to last for 3 years. Company A estimates that it will cost $500,000 to dismantle and dispose of the device after 3 years. The company's capital rate is 10%, and there are no regulatory requirements for the disposal of the device. What should be the cost of the device?

A. $3,000,000

B. $3,200,000

C. $3,700,000

D. $3,575,657

答案：B
Answer: B

Figure 65: An "accounting" example. The English translation is given beneath the corresponding Chinese text and is highlighted in blue.

問題：建築物發生閃燃（Flashover）現象，下列敘述何者錯誤？
Question: Which of the following statements is incorrect about the flashover phenomenon in buildings?

A. 延遲建築物火災到達閃燃階段，為最重要的避難對策

A. Delaying the arrival of a building fire to the flashover stage is the most important evacuation strategy.

B. 閃燃現象通常發生在建築物火災的成長期

B. Flashover phenomena usually occur during the growth phase of a building fire.

C. 發生閃燃現象時，居室內的人生存機會低

C. When a flashover occurs, the chances of survival for people in the room are low.

D. 同種類等面積之裝潢材料中，壁面影響閃燃時間（F.O.T.）最大

D. In the same type of decorative material with equal area, the wall surface has the greatest influence on the Flashover Time (F.O.T.).

答案：D
Answer: D

Figure 66: A "fire science" example. The English translation is given beneath the corresponding Chinese text and is highlighted in blue.

問題：患者右眼前方放置紅色片（red lens test），看到紅光點在他的右下方，則患者有：

Question: If the patient has a red lens placed in front of the right eye (red lens test), and sees a red light point in the lower right, the patient has:

A. 右眼下內隱斜位

A. Right eye's lower internal latent strabismus

B. 右眼上外隱斜位

B. Right eye's upper external latent strabismus

C. 左眼上內隱斜位

C. Left eye's upper internal latent strabismus

D. 右眼上內隱斜位

D. Right eye's upper internal latent strabismus

答案：D
Answer: D

Figure 67: An "optometry" example. The English translation is given beneath the corresponding Chinese text and is highlighted in blue.

問題：甲以自己為要保人與被保險人，向A 人壽保險公司（下稱A 公司）投保醫療險，並加保新冠肺炎法定傳染病醫療險附約，住院每日理賠金額為新臺幣5000 元，並預繳第一期保費。甲於投保後，A 公司核保前確診新冠肺炎住院治療，甲於6 日後康復出院。下列敘述何者正確？

Question: After ensuring himself as the insurer and the insured with A Life Insurance Company (hereinafter referred to as A Company), and underwriting medical insurance for coronavirus disease, with a hospitalization daily claim amount of NTD 5,000 and prepaying the first premium, the patient was diagnosed with COVID-19 and hospitalized for treatment before A Company underwrote the insurance, and was discharged from the hospital 6 days later. Which of the following statements is true?

A. A 公司同意承保時，保險期間溯自預收第一期保險費金額時開始

A. When A Company agrees to underwrite the insurance, the insurance period is retroactive from the time the first insurance premium amount is prepaid.

B. A 公司得拒絕承保

B. A Company can refuse to underwrite the insurance.

C. A 公司應理賠甲新臺幣3 萬元

C. A Company should compensate him NTD 30,000.

D. A 公司得以未經核保，主張契約效力未定

D. A Company can claim that the contract is not yet effective without underwriting.

答案：C
Answer: C

Figure 68: An "insurance studies" example. The English translation is given beneath the corresponding Chinese text and is highlighted in blue.

問題：下列常用口服adrenocorticoids藥，何者脂溶性最大？
Question: Which of the following commonly used oral adrenocorticoids has the greatest lipid solubility?

A. prednisolone

B. triamcinolone

C. dexamethasone

D. hydrocortisone

答案：C
Answer: C

Figure 69: A "pharmacology" example. The English translation is given beneath the corresponding Chinese text and is highlighted in blue.

問題：甲公司聯合製程產出A、B兩種產品，聯合成本為$48,000。A產品共計5,000單位，接受分攤之聯合成本金額為$31,200。A產品在分離點可用每單位$24出售，亦可額外投入成本$15,000繼續加工，並以每單位$26出售。關於A產品之敘述，下列何者正確？Question: Company A produces two products, A and B, through a joint process with a joint cost of $48,000. Product A is produced in 5,000 units, and the cost allocated from the joint cost is $31,200. Product A can be sold at $24 per unit at the separation point, or it can be processed further with an additional cost of $15,000 and sold at $26 per unit. Which of the following statements about product A is correct?

A. 繼續加工並出售的利潤較少，少$36,200

A. The profit from further processing and selling is less, $36,200 less

B. 繼續加工並出售的利潤較多，多$115,000

B. The profit from further processing and selling is more, $115,000 more

C. 繼續加工並出售的利潤較少，少$5,000

C. The profit from further processing and selling is less, $5,000 less

D. 繼續加工並出售的利潤較多，多$26,200

D. The profit from further processing and selling is more, $26,200 more

答案：C
Answer: C

Figure 70: A "management accounting" example. The English translation is given beneath the corresponding Chinese text and is highlighted in blue.

問題：下列有關木材防腐方法的敘述，何者錯誤？
Question: Which of the following statements about wood preservation methods is incorrect?

A. 熱水浸漬法可防止白蟻蟲害

A. Hot water immersion can prevent termite damage.

B. 炭化之表面處理可達防腐效果

B. Surface treatment by carbonization can achieve anti-corrosion effect.

C. 加壓注入焦油或防腐藥劑，其防腐效力最佳

C. Pressure injection of coal tar or preservatives, has the best anti-corrosion effect.

D. 常壓熱冷浸漬藥劑防腐法可達防腐效果

D. The normal pressure hot and cold immersion agent anti-corrosion method can achieve the anti-corrosion effect.

答案：A
Answer: A

Figure 71: An "agriculture" example. The English translation is given beneath the corresponding Chinese text and is highlighted in blue.

問題：依部頒「電子公文發文附件使用ODF格式管制稽核作法」規定，以下何者非ODF格式之電子檔案副檔名?
Question: According to the "Regulations on the Use of ODF Format for Electronic Document Attachments", which of the following is not a ODF format electronic file extension?

A. doc

B. docx

C. pptx

D. 以上皆非。

D. None of the above.

答案：D
Answer: D

Figure 72: An "official document management" example. The English translation is given beneath the corresponding Chinese text and is highlighted in blue.

問題：哪一比率常被認為是影響股票報酬率的基本面因素？
甲.本益比；
乙.淨值市價比；
丙.公司規模大小
Question: Which ratio is often considered a fundamental factor affecting stock return rate? Price-Earnings Ratio of A; Net Asset Value Ratio of B; Company C Size

A. 僅甲、丙

A. Only A, C

B. 僅甲、乙

B. Only A, B

C. 僅乙、丙

C. Only B, C

D. 甲、乙、丙

D. A, B, C

答案：D
Answer: D

Figure 73: A "financial analysis" example. The English translation is given beneath the corresponding Chinese text and is highlighted in blue.

問題：消費者市場包含了許多不同角色的參與者。請問其中哪一種角色是指在消費的過程中，提出意見且左右購買決策的人？
Question: The consumer market includes many different participants. Which role referred as the one who gives opinion and influences the purchase decision during the consumption process?

A. 提議者

A. Proposer

B. 決策者

B. Decider

C. 影響者

C. Influencer

D. 購買者

D. Buyer

答案：C
Answer: C

Figure 74: A "marketing management" example. The English translation is given beneath the corresponding Chinese text and is highlighted in blue.

問題：促進變革的外部驅動力不包括下列何者？
Question: Which of the following is not an external driving force for promoting change?

A. 技術革新

A. Technological innovation

B. 經濟環境改變

B. Change in economic environment

C. 顧客需求改變

C. Change in customer demands

D. 員工績效評估方式改變

D. Change in employee performance evaluation method

答案：C
Answer: C

Figure 75: A "business management" example. The English translation is given beneath the corresponding Chinese text and is highlighted in blue.

問題：保險人對於要保人或被保險人，為避免或減輕損害之必要行為所生之救助費用，應如何償還？
Question: How should the insurer reimburse the salvage expenses necessary for the policyholder or the insured to avoid or mitigate damage?

A. 除契約另有約定外，保險人依保險標的價值償還救助費用

A. Except as otherwise specified in the contract, the insurer reimburses the rescue costs according to the value of the insured subject.

B. 保險人須償還數額與賠償金額，但應以保險金額為限

B. The insurer must reimburse the amount and compensation amount, but should be limited to the insurance amount.

C. 保險人償還數額與賠償金額，合計雖超過保險金額，仍應償還救助費用

C. The insurer reimburses the amount and compensation amount, even if the total exceeds the insurance amount, the rescue costs should still be reimbursed.

D. 除契約另有約定外，保險人不須償還救助費用

D. Except as otherwise specified in the contract, the insurer does not need to reimburse the rescue costs.

答案：C
Answer: C

Figure 76: A "finance banking" example. The English translation is given beneath the corresponding Chinese text and is highlighted in blue.

## J   Evaluation Environment

Our evaluation environment differentiates between open-weight and closed-source models. Text generation inference is conducted on a workstation equipped with eight A6000 GPUs for open-weight models running in bfloat16. In the case of closed-source models, we utilize the Python package provided by their vendor.

