# OpenReview forum: "TMMLU+: An Improved Traditional Chinese Evaluation Suite for Foundation Models"
_colmweb.org/COLM/2024/Conference — COLM_

### Official Review · Reviewer_Zs62 · 2024-05-10

**Rating:** 6
**Confidence:** 3
**Ethics Flag:** 1

**Summary:**

This paper introduces TMMLU+, an improved benchmark dataset for evaluating the language understanding capabilities of large language models (LLMs) in Traditional Chinese. TMMLU+ consists of 22,690 multi-choice questions across 66 subjects, ranging from elementary to professional levels. The authors conduct experiments on 26 open-weight language models and several closed-source models, comparing their performance against human test-takers. The findings reveal that Traditional Chinese models still lag behind their Simplified Chinese counterparts, and current LLMs fall short of human performance in average scores, particularly in social science and humanities subjects.

**Reasons To Accept:**

- The paper introduces TMMLU+, improving the dataset size and subject diversity of its predecessor.
- The paper provides a comprehensive evaluation of a wide range of language models, including both open-weight and closed-source models, allowing for a thorough comparison of their performance on Traditional Chinese language understanding tasks.

**Reasons To Reject:**

- While the paper compares the performance of Traditional and Simplified Chinese language models, it does not provide the underlying reasons for the performance gap between the two. Further investigation into the linguistic and cultural differences between Traditional and Simplified Chinese could provide more insights. (especially for readers who are not familiar with the Chinese.)
- The process of creating the CoT demonstration has been overly simplified, and there are concerns about whether the quality has been adequately ensured. Additionally, there is no explanation for why the performance of CoT is lower compared to direct methods.

---

> ### Author Rebuttal · Authors · 2024-05-30
>
> 1. Here are some potential reasons for further investigation:
>
> Notation: TC = Traditional Chinese, SC = Simplified Chinese
>
> **One-to-Many:** Simplification in SC has led to certain characters having multiple meanings, distinct from TC. The issue is exacerbated by shared tokens between the two writing systems, impacting interpretation and vocabulary, and potentially leading to confusion.
>
> | Common Token | TC Usage | SC Usage |
> |---|---|---|
> | 云 | 雲(cloud) | 云(cloud/say) |
> |  | 云(say) |  |
> | 丑 | 醜(ugly) | 丑(ugly/1am-3am) |
> |  | 丑(1am-3am) |  |
>
> **Terminology Difference:** Transitioning between TC and SC requires nuanced translation beyond character substitution, involving differences in character forms, vocabulary, and usage, and necessitates an understanding of cultural and historical contexts, as shown in examples from TMMLU+ and C-Eval.
>
> | TC | SC | 1 to 1 Character Translation | Meaning |
> |---|---|---|---|
> | 記憶體 | 内存 | 记忆体 | memory |
> | 計程車 | 出租车 | 计程车 | taxi |
>
> **Regional Linguistic Variations:** Language evolution, influenced by popular culture and regional traits, results in unique characteristics and vocabulary reflecting geographical and cultural differences. For instance, TC incorporates elements from Taiwanese Hokkien, introducing new terms like the informal expression "母湯" (It's not okay!) and "囡仔" (little kids), commonly used in casual conversations.
>
> 2. Our CoT reasonings were created using the methodology from C-Eval Appendix C, which utilizes GPT-4 for initial reasoning followed by human review.
>
> We also reviewed reasoning from gpt-3.5-turbo, finding it capable of generating steps, with our parser parsing the final answer successfully. However, some CoT answers were incorrect resulting in a performance drop. This decline is evident in Table 5 of C-Eval [1], with GPT-3.5-turbo's average score decreasing from 54.4 to 50.0.
>
> We attribute this drop to insufficient relevance among bridging objects, detailed in Section 5 [2]. As shown in the CoT example [3], GPT-3.5-turbo incorrectly assumed fewer sediments result in lighter water. Instead, the correct reasoning is that fewer sediments enable light to penetrate deeper, resulting in darker water due to its depth.
>
> [1] Huang, Yuzhen et al. C-Eval: A Multi-Level Multi-Discipline Chinese Evaluation Suite for Foundation Models. NeurIPS 2023
>
> [2] Wang, Boshi et al. Towards Understanding Chain-of-Thought Prompting: An Empirical Study of What Matters. ACL 2023
>
> [3] https://pastebin.com/ypdXPtMv

---

### Official Review · Reviewer_bAwS · 2024-05-10

**Rating:** 7
**Confidence:** 3
**Ethics Flag:** 1

**Summary:**

The paper introduces TMMLU+, an enhanced Traditional Chinese Evaluation Suite designed to assess the capabilities of LLMs in traditional Chinese. The benchmark is divided into four categories with a total of 22,690 multi-choice questions. The paper conducts a thorough evaluation of several representative LLMs, such as close-source LLMs LLMs (e.g., chatGPT, GPT4) and many open-weight LLMs, including those tailored for simplified and traditional Chinese.

**Questions To Authors:**

1. Appendix E lists many LLMs. Why Table 2 doesnot report the performance of all LLMs? Additionally, Figure 4 includes additional LLMs not featured in Table 2. In abstract it says 26 open-weight Chinese LLMs, it seems not ture.

2. How are examples chosen for the development and validation sets? Table 2 observes decline in performance for many LLMs with few-shot prompting. Is there a potential correlation with the examples in the development set?

3. LLMs exhibit significantly reduced performance under the CoT setting compared to without. What are the reasons behind?

4. How much money you spent on GPT family models?

**Reasons To Accept:**

1. The paper presents a traditional Chinese benchmark, TMMLU+, covering 66 subjects with a total of 22,690 questions.

2. The paper provides an in-depth evaluation of many representative LLMs, offering valuable insights into their traditional Chinese performance.

**Reasons To Reject:**

1. The benchmark in this work has not well distinguished itself among the many existing benchmarks for Chinese. It shares similarities with benchmarks tailored for simplified Chinese, such as C-Eval.

2. The dataset's license should be explicitly specified in the paper.

---

> ### Author Rebuttal · Authors · 2024-05-30
>
> * Reply to Rejection:
>
> 1. Notation: TC = Traditional Chinese, SC = Simplified Chinese
>
> We cover regional topics such as taxes, laws, insurance, accounting, finance, humanities, geography, TTQA and Taiwan-specific agricultural practices. Moreover, TTQA explores Taiwanese culture, geography, and history. While overlapping with C-Eval in many STEM fields, our terminology differs. For instance, key computer science terms use different characters. Translation tools like OpenCC may map traditional characters to simplified ones, but they don't accurately represent those used in SC.
> |TC|SC|1-to-1 Character Translation|Meaning|
> |-|-|-|-|
> |記憶體|内存|记忆体|memory|
> |計程車|出租车|计程车|taxi|
>
> Nuances also appear at the character level in both languages.
> |Char|TC Usage|SC Usage|
> |-|-|-|
> |丑|醜(ugly)|丑(ugly/1am-3am)|
> ||丑(1am-3am)||
> |云|雲(cloud)|云(cloud/say)|
> ||云(say)||
>
> 2. MIT license
>
> * Reply to Questions:
>
> 1. We included only some LLMs in Table 2 to avoid overwhelming the reader. We found a miscount by -3 models and will correct this in the revision.
>
> 2. We shuffle question sets for each subject to ensure a balanced mix, choosing the first 5 questions for our development set and dividing the rest between validation and testing (1:9 ratio).
>
> In Table 2, 10 out of 17 LLMs excel with 5-shot prompting, while only 5 models excel with 0-shot. The lower effectiveness of 5-shot prompting in some models may be due to a limited context size, complicating focus on relevant questions, especially in smaller or less optimized models, as shown by declines in C-Eval[1]. We also review the development set to ensure that the topics are covered in the testing sets.
>
> 3. We reviewed GPT-3.5-turbo's reasoning, finding it capable of generating steps with the final answers successfully parsed. However, some CoT answers were incorrect, resulting in a performance drop. This decline is evident in C-Eval[1], where GPT-3.5-turbo's average score decreased from 54.4 to 50.0.
>
> We attribute this drop to insufficient relevance among bridging objects, as discussed in Section 5[2]. As in  CoT example[3], GPT-3.5-turbo incorrectly assumed that fewer sediments result in lighter water, leading to the selection of the wrong answer.
>
> 4. USD 1341.63
>
> [1] Huang, Yuzhen et al. C-Eval: A Multi-Level Multi-Discipline Chinese Evaluation Suite for Foundation Models. NeurIPS 2023
>
> [2] Wang, Boshi et al. Towards Understanding Chain-of-Thought Prompting: An Empirical Study of What Matters. ACL 2023
>
> [3] https://pastebin.com/ypdXPtMv

---

> > ### Comment · Reviewer_bAwS · 2024-06-04
> >
> > Thank you for the clarifications. After considering the authors' response, I've modified my score accordingly.

---

> > > ### Author Response · Authors · 2024-06-06
> > > **Thanks for pointing out weak points**
> > >
> > > We are grateful for bAwS's keen observations on sections needing improvement, which will guide changes to our revisions.

---

### Official Review · Reviewer_8bdQ · 2024-05-11

**Rating:** 7
**Confidence:** 4
**Ethics Flag:** 1

**Summary:**

This paper introduces TMMLU+, an expanded and enhanced benchmark suite designed to assess language model performance on Traditional Chinese language understanding tasks. Compared to its predecessor, TMMLU, it offers a more comprehensive array of multiple-choice questions across a wider range of subjects, with improved balance in subject distribution. The authors conduct extensive analyses examining various model features, such as performance differences between simplified and traditional characters, the use of Chain-of-Thought (CoT), tokenizer vocabulary size, models and human performance.

**Questions To Authors:**

In Figure 8, the English prompt is not correctly translated.

**Reasons To Accept:**

1. TMMLU+ significantly broadens the scope of its predecessor. It also includes questions specific to Taiwanese culture, like Taiwanese Hokkien and aboriginal cultures.
2. The authors provide a thorough analysis of different model features, aiming to explain the reasons behind varying model performances.
3. The paper offers interesting and inspiring findings, such as the observation that models perform better than human test-takers on difficult questions.

**Reasons To Reject:**

The benchmark primarily focuses on knowledge-based questions, leading to significant overlap in terminology between Mainland Mandarin and Taiwanese Mandarin. While some questions reflect real language usage differences (e.g., Figure 8 in Appendix D and the Taiwanese Hokkien questions), they constitute a small portion of the overall dataset.

---

> ### Author Rebuttal · Authors · 2024-05-30
>
> Thanks 8bdQ for the detailed overview and concern.
>
> > The benchmark primarily focuses on knowledge-based questions, leading to significant overlap in terminology between Mainland Mandarin and Taiwanese Mandarin. While some questions reflect real language usage differences (e.g., Figure 8 in Appendix D and the Taiwanese Hokkien questions), they constitute a small portion of the overall dataset.
>
> Although the underlying knowledge (rules of physics and parts of history and culture) overlaps, our questions also focus on regional knowledge, including legal laws, specialized agricultural practices unique to Taiwan, insurance, accounting rules, finance acts, humanities, and geography, which are key aspects of daily life in Taiwan. Additionally, we included questions related to indigenous people in Taiwan. If any large language models are to be deployed in this region, having a benchmark to ensure the model has a good understanding of the society is important.
>
> Our benchmark specifically addresses the terminology differences between Traditional and Simplified Chinese, ensuring that models can accurately understand and respond to questions in the context of Taiwan. The terminology used in Traditional Chinese often differs from that in Simplified Chinese, reflecting variations in character forms, vocabulary, and usage. For instance, consider the computer science subject in TMMLU+ and the computer architecture section in C-Eval.
>
> Both questions inquire about the "four important stages of computer development" but use different terms for "computer" (`電腦` and `电子计算机`).
>
> - The 100th question from the computer science subject in TMMLU+ asks:
>
> `電腦的組成元件歷經四個重要階段，下列的發展順序（由先至後）何者正確？`
>
> - The 81st question from the computer architecture section in C-Eval asks:
>
> `电子计算机的发展已经经历了4代，这4代计算机的主要元件分别是____。`
>
> Hence, our dataset could serve as a future work to compare the differences between Taiwanese Mandarin and Mainland Mandarin evaluations (C-Eval) when pre-trained on a different mix of character sources for pretraining.
>
> > In Figure 8, the English prompt is not correctly translated.
>
> Noted, we thank the reviewer for spotting this error and will include the corrected translation prompt in our latest version.

---

### Decision · Program_Chairs · 2024-07-10

**Decision:**

Accept

**Comment:**

The paper presents a traditional Chinese benchmark, TMMLU+, covering 66 subjects with a total of 22,690 questions, along with an in-depth evaluation of many representative LLMs, offering valuable insights into their traditional Chinese performance. The reviewers were all supportive; this will be a very valuable benchmark.